# Immunophenotypic correlates of sustained MRD negativity in patients with multiple myeloma

David G. Coffey [1,2] ✉, Francesco Maura [1], Edgar Gonzalez-Kozlova [3], J. Javier Diaz-Mejia [4], Ping Luo [4], Yong Zhang[5], Yuexin Xu[2], Edus H. Warren [2], Travis Dawson [3], Brian Lee[3], Hui Xie[3], Eric Smith [6], Amanda Ciardiello[7], Hearn J. Cho [3,8], Adeeb Rahman[3], Seunghee Kim-Schulze[3], Benjamin Diamond [1], Alexander Lesokhin[7,11], Dickran Kazandjian [1,11], Trevor J. Pugh[4,9,10,11], Damian J. Green[2,11], Sacha Gnjatic [3,11] & Ola Landgren [1,11] ✉

The role of the immune microenvironment in maintaining disease remission in patients with multiple myeloma (MM) is not well understood. In this study, we comprehensively profile the immune system in patients with newly diagnosed MM receiving continuous lenalidomide maintenance therapy with the aim of discovering correlates of long-term treatment response. Leveraging single-cell RNA sequencing and T cell receptor β sequencing of the peripheral blood and CyTOF mass cytometry of the bone marrow, we longitudinally characterize the immune landscape in 23 patients before and one year after lenalidomide exposure. We compare patients achieving sustained minimal residual disease (MRD) negativity to patients who never achieved or were unable to maintain MRD negativity. We observe that the composition of the immune micro-environment in both the blood and the marrow varied substantially according to both MRD negative status and history of autologous stem cell transplant, supporting the hypothesis that the immune microenvironment influences the depth and duration of treatment response.

The overall survival of patients diagnosed with multiple myeloma (MM) has significantly improved in recent decades[1]. The lack of established curative therapies however leads to relapse and consequently, the current treatment paradigm is focused on using maintenance therapy to prolong and improve initial treatment response[2]. Large, randomized clinical trials have demonstrated improved progression-free survival (PFS) and overall survival in patients with newly diagnosed and relapsed MM receiving the immunomodulatory drug (IMiD) lenalidomide as maintenance therapy[3]. However, the optimal duration of maintenance remains controversial[4].

Members of our study team recently published the results of phase II clinical trial investigating the dynamics of minimal residual

[1]Division of Myeloma, Sylvester Comprehensive Cancer Center, University of Miami, Miami, FL, USA. [2]Clinical Research Division, Fred Hutchinson Cancer Research Center, Seattle, WA, USA. [3]Icahn School of Medicine at Mount Sinai, New York, NY, USA. [4]Princess Margaret Cancer Centre, University Health Network, Toronto, ON, Canada. [5]Office of Oncologic Diseases, Center for Drug Evaluation and Research, U.S. Food and Drug Administration, Silver Spring, MD, USA. [6]Dana-Farber Cancer Institute, Boston, MA, USA. [7]Myeloma Service, Department of Medicine, Memorial Sloan Kettering Cancer Center, New York, NY, USA. [8]Multiple Myeloma Research Foundation, Norwalk, USA. [9]Ontario Institute for Cancer Research, Toronto, ON, Canada. [10]Department of Medical Biophysics, University of Toronto, Toronto, ON, Canada. [11]These authors contributed equally: Alexander Lesokhin, Dickran Kazandjian, Trevor J. Pugh, Damian J. Green, Sacha Gnjatic, Ola Landgren. ✉e-mail: davidcoffey@miami.edu; col15@miami.edu

disease (MRD) during lenalidomide maintenance therapy for patients with newly diagnosed MM following unrestricted induction therapy[5]. In this trial, it was demonstrated that achieving sustained MRD negativity, defined as two consecutive negative measurements at least 1 year apart during maintenance therapy, is strongly associated with prolonged PFS. It was found that compared to patients with sustained MRD negativity, patients who lost MRD negativity after one year of maintenance were 14 times more likely to progress than those who sustained MRD negativity. In fact, patients who sustained MRD negativity for 2 years ($n = 34$) had no recorded disease progression at a median of 19.8 months (95% CI 15.8–22.3) past the 2-year maintenance landmark.

Based on these clinical observations, it has been proposed that there may be a subset of patients who can attain a functional cure, demonstrated by persistently absent disease at the lowest levels of detection, and these individuals may benefit from cessation of maintenance therapy[6]. However, the challenge remains in prospectively identifying patients appropriate for treatment de-escalation. While our group and others have uncovered genomic alterations responsible for disease initiation and progression[7–9], characterizing the malignant population of cells to define the molecular underpinnings of deep clinical response is not currently feasible in patients who are MRD negative since the tumor cells are below the limit of detection. While some studies have shown clear evidence of genomic and transcriptional changes in the tumor cells at the time of relapse, no consistent mechanisms of lenalidomide resistance have been uncovered[10–12]. Elucidating such mechanisms, however, may enable a risk-adapted approach to maintenance therapy.

Since one of the mechanisms of action of lenalidomide is to enhance cell-mediated immunity by stimulating proliferation and activation of T cells and natural killer (NK) cells[13], we hypothesized that the duration of the clinical response may also be driven by features of the immune microenvironment. In support of this, a recent study focusing on myeloma precursor conditions, monoclonal gammopathy of undetermined significance (MGUS) and smoldering MM (SMM), found substantial transcriptional and compositional changes in the bone marrow microenvironment during progression from a healthy state, to precursor MM, and then active disease[14]. Therefore, we conducted a comprehensive investigation of the immune landscape in patients with newly diagnosed MM receiving lenalidomide as maintenance therapy. Using

longitudinally collected bone marrow and peripheral blood samples before and during lenalidomide treatment, we performed an exploratory, secondary analysis of our previously published clinical trial to elucidate the roles of the immune cell subsets and T-cell repertoire. We leveraged single-cell RNA sequencing (scRNAseq) coupled with V(D)J sequencing of peripheral blood mononuclear cells (PBMCs), CyTOF mass cytometry of bone marrow mononuclear cells (BMMCs), and T-cell receptor β (TCR β) repertoire analysis of PBMCs to uncover features within the immune microenvironment that distinguish patients achieving sustained MRD negativity from those who lost or never attained an MRD-negative state during the first year of maintenance therapy.

In this work, we show that patients with sustained MRD negativity had an increase in circulating helper T cells and reduced exhausted T cells and suppressive regulatory T cells (Treg cells). In addition, we found that the frequency of immune cell subsets in patients with sustained MRD negativity and no prior history of autologous stem cell transplant most resembled healthy donors indicating that the immune systems of patients achieving a deep remission may recover to a healthy state.

## Results

### Characteristics of the study population

To uncover features of long-term treatment response, we comprehensively profiled the immune microenvironment of 23 patients before and approximately one year after lenalidomide maintenance therapy. We compared 12 patients with sustained MRD negativity during the first year of therapy to 11 patients who lost or were unable to attain an MRD-negative state (Fig. 1a, b). Sustained MRD negativity was defined as achieving two consecutive MRD-negative measurements one year apart while non-sustained MRD negativity included all other patients, including those who never achieved MRD negativity. In the non-sustained MRD-negative group, there were nine patients who had persistent MRD positivity from the start of maintenance and two patients who converted from MRD negative to positive within the first year of maintenance. The median PFS for all patients in this correlative study was 22 months (Fig. 1c). MRD-negative status at one year was strongly associated with PFS as 100% of patients with non-sustained MRD negativity progressed within 24 months, whereas 0% progressed during that same period if they had achieved sustained MRD negativity (log-rank $P$ value = $4.4 \times 10^{-16}$, Fig. 1d).

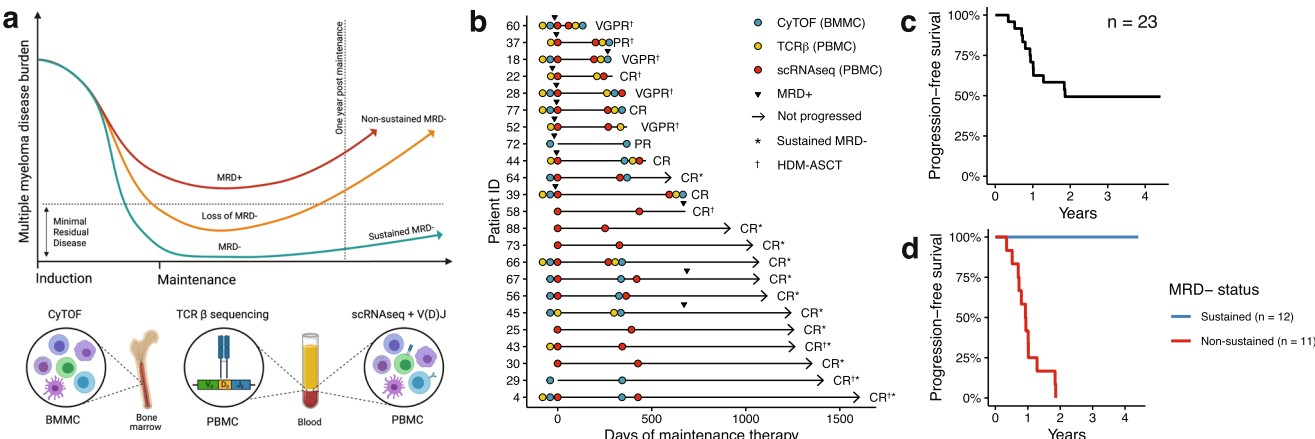

**Fig. 1 | Study design and treatment outcomes. a** Immune profiling was performed on blood and bone marrow collected from 23 patients with multiple myeloma receiving lenalidomide as maintenance therapy (created with BioRender.com). **b** Swimmer plot showing duration and depth of response to lenalidomide in relation to the sample collection time points and immune assays performed. **c** Kaplan–Meier survival analysis of progression-free survival from the start of the initiation of maintenance therapy for all patients and **d** stratified according to the minimal residual disease (MRD) negative status. PR partial response, VGPR very good partial response, CR complete response, TCR T-cell receptor, BMMC bone marrow mononuclear cells, PBMC peripheral blood mononuclear cells. Source data are provided as a Supplementary Data file.

**Table 1 | Characteristics of study patients (n = 23)**

| | MRD- status | | |
| --- | --- | --- | --- |
| | Non-sustained | Sustained | P |
| *n* (%) | 11 (47.8) | 12 (52.2) | |
| Age (mean (SD)) | 67.6 (10.7) | 64.8 (10.5) | 0.534 |
| **Sex (%)** | | | |
| Female | 2 (18.2) | 1 (8.3) | 0.590 |
| Male | 9 (81.8) | 11 (91.7) | |
| **Stage (%)** | | | |
| ISS I | 4 (36.4) | 8 (66.7) | 0.425 |
| ISS II | 4 (36.4) | 3 (25.0) | |
| ISS III | 2 (18.2) | 0 (0.0) | |
| **Induction** | | | |
| IMiD | 5 (45.5) | 11 (91.7) | 0.027 |
| No IMiD | 6 (54.5) | 1 (8.3) | |
| Carfilzomib | 2 (18.2) | 8 (66.7) | 0.036 |
| No Carfilzomib | 9 (81.8) | 4 (33.3) | |
| Bortezomib | 10 (90.9) | 4 (33.3) | 0.009 |
| No Bortezomib | 1 (9.1) | 8 (66.7) | |
| Prior HDM-ASCT | 7 (63.6) | 3 (25.0) | 0.100 |
| No prior HDM-ASCT | 4 (36.4) | 9 (75.0) | |
| **Cytogenetics (%)** | | | |
| High-risk | 4 (36.4) | 0 (0.0) | 0.037 |
| Standard-risk | 7 (63.6) | 12 (100.0) | |

Two-sided *P* values for Fisher's exact test are shown for categorical variables and *T* tests for numerical variables. Source data are provided as a Supplementary Data file.
ISS, International Staging System; high-risk cytogenetics defined as del17p, t(4;14), t(14;16), 1q+.

Characteristics of the study population according to their MRD-negative status are shown in Table 1. Patients with sustained MRD negativity were significantly more likely to lack high-risk cytogenetics and have received an immunomodulatory drug and/or carfilzomib during induction therapy. The choice of induction therapy was variable with 23 (100%) patients receiving a proteasome inhibitor, 16 (70%) receiving an IMiD, 5 (22%) receiving an alkylator, and 23 (100%) receiving dexamethasone (Supplementary Data 1).

**Immune profiling of PBMCs by single-cell RNA sequencing**
scRNAseq was used to characterize the cellular composition and phenotypic changes of PBMCs immediately before lenalidomide maintenance (baseline) and again after 1 year of treatment (follow-up). After quality filtering, a total 95,006 PBMCs were analyzed (median, 2027 cells/sample; range, 311–6601 cells/sample). Automated reference-based mapping using Azimuth from the Seurat package classified 21 types of leukocytes. Dimensionality reduction using uniform manifold approximation and projection (UMAP) revealed three primary clusters corresponding to T/NK cells, B cells, and myeloid cells (Fig. 2a and Supplementary Fig. S1). Genes that are known to be associated with each identified cell type were found to be specifically expressed (Supplementary Fig. 2). V(D)J sequencing confirmed the presence of the correct antigen receptor expressed by T and B lymphocytes (Fig. 2b). Clonal expansion was most common in terminal effector CD8⁺ T cells and not present in B-cell subsets (Supplementary Fig. 3).

To control for a history of HDM-ASCT, which was more prevalent among patients with non-sustained MRD negativity, sustained versus non-sustained MRD-negative patients were compared separately based on their transplant history (Supplementary Fig. 4). On average, terminal effector CD8⁺ T cells were the most frequent cell type at baseline and follow-up for patients with sustained MRD negativity and

prior HDM-ASCT (Fig. 2c) resulting in a reduced CD4/CD8 ratio (Supplementary Fig. 5 and Supplementary Data 2). In contrast, a higher CD4/CD8 ratio was observed for all other comparisons, especially in patients with sustained MRD negativity without prior HDM-ASCT, in whom CD4⁺ central memory and naive CD4⁺ T cells were the most abundant lymphocyte. Naive B cells were the most common B-cell phenotype for all comparisons and time points (Fig. 2d). Circulating plasmablasts were detected at very low frequency and were higher in patients with non-sustained MRD negativity, especially at the follow-up time point, potentially reflecting disease progression. Classical monocytes were the most common myeloid cell type and were reduced in patients who had a prior HDM-ASCT (Fig. 2e).

Differential abundance analysis comparing patients by their MRD-negative status and HDM-ASCT history at baseline and follow-up revealed that among patients with sustained MRD negativity, naive CD8⁺ T cells were higher at baseline and naive CD4⁺ T cells were higher at follow-up (Fig. 2f). The same finding was also detected in patients without prior HDM-ASCT (Fig. 2g). Comparison of the percent change in frequency of cell types from baseline to following revealed that Treg cells significantly decreased in patients with sustained MRD negativity (Wilcoxon rank-sum test *P* = 0.026) while MAIT cells (Wilcoxon rank-sum test *P* = 0.018), naive CD8⁺ T cells (Wilcoxon rank-sum test *P* = 0.004), and Treg cells (Wilcoxon rank-sum test *P* = 0.001) were significantly increased in patients with prior HDM-ASCT (Fig. 2h).

To reveal differences in gene expression within cell types across MRD-negative subgroups, we performed a sub-clustering analysis (Supplementary Figs. 6–11). This was combined with a differential gene expression analysis to uncover genes specific to these subpopulations. We found that for some cell types, there are subpopulations of cells that cluster separately according to their MRD-negative status[15]. In addition, we detected genes that are differentially expressed at baseline and during follow-up time points that are associated with sustained MRD negativity.

To determine the state of exhaustion among the various circulating T-cell subsets, we identified terminal effector memory T cells ($T_{EM}$) classified by Azimuth which are expressing any one of the following genes associated with T-cell exhaustion in human cancer: Lymphocyte Activating 3 (*LAG3*), layilin (LAYN), Programmed cell death protein 1 (*PDCD1*), T-cell immunoreceptor with Ig and ITIM domains (*TIGIT*), Cytotoxic T-Lymphocyte Associated Protein 4 (*CTLA4*), Thymocyte Selection Associated High Mobility Group Box (TOX), Hepatitis A Virus Cellular Receptor 2 (HAVCR2), Integrin Subunit Alpha E (ITGAE), C-X-C Motif Chemokine Ligand 13 (CXCL13) (Supplementary Figs. S12–S13)[16]. Differential abundance analysis identified that the percentage of $T_{EM}$ cells expressing *TOX* at follow-up was significantly higher in patients with non-sustained MRD negativity and no prior history of ASCT (Wilcoxon rank-sum test *P* = 0.048, Fig. 3a Supplementary Data 3). We next compared the frequency of *TOX* expressing $T_{EM}$ cells over time and observed a significant trend toward increasing cell frequency among patients with non-sustained MRD negativity and prior HDM-ASCT (Wilcoxon rank-sum test *P* = 0.011, Fig. 3b). A similar, but non-significant trend was also observed among patients with non-sustained MRD negativity and no prior HDM-ASCT. Importantly, the frequency of *TOX* + $T_{EM}$ cells did not appear to change in patients with sustained MRD negativity. Given the small sample size in our study, we advise that the statistical findings be interpreted with caution and verified in larger studies.

Since we simultaneously sequenced the T-cell receptor in combination with its gene expression, we compared the diversity of the CD4⁺ and CD8⁺ T-cell repertoire across time points and patient groups. A paired alpha-beta T-cell receptor was detected on 50% of the 50,625 T cells classified (Supplementary Fig. 14 and Supplementary Data 4). As expected, we observed a larger proportion of the CD8⁺ T-cell repertoire to be clonal (higher Gini index) compared to the CD4⁺ T-cell repertoire (Fig. 3c and Supplementary Data 5). We did not observe a

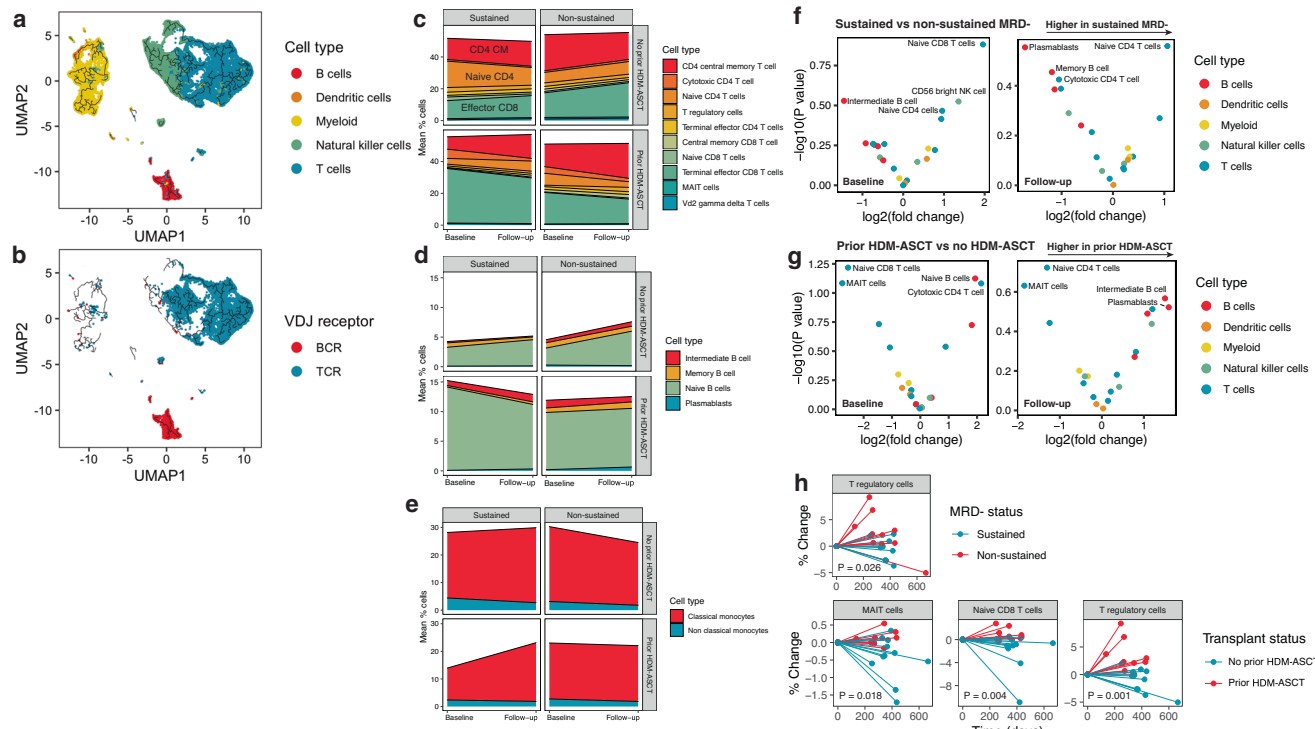

**Fig. 2 | scRNAseq of PBMC reveals differences in immune cell frequency according to minimal residual disease (MRD)-negative status and transplant history. a** Aggregate UMAP and trajectory analysis of all cells analyzed color-coded by cell type and **b** presence of B-cell receptor (BCR) or T-cell receptor (TCR) by V(D)J sequencing. **c** Area plot showing the mean frequency of T cells, **d** B cells, and **e** myeloid cells at baseline and follow-up according to the patients' MRD-negative status and transplant history. **f** Volcano plot comparing differentially abundant circulating immune cells between patients achieving sustained and non-sustained MRD negativity at baseline (left) and follow-up (right). **g** Volcano plot comparing differentially abundant circulating immune cells between patients with prior

history of transplant or not at baseline (left) and follow-up (right). **h** Spider plot showing the percent change in cell frequency between baseline and follow-up time points according to the patients' MRD-negative status (top row) and transplant history (bottom row). Only cell types that are significantly increased or decreased over time (Wilcoxon rank-sum test $P < 0.05$) are shown. $N = 20$ sustained MRD- (16 no prior HDM-ASCT, 4 prior HDM-ASCT) and 20 non-sustained MRD- (6 no prior HDM-ASCT, 14 prior HDM-ASCT) patient samples. Source data are provided as a Supplementary Data file. HDM-ASCT high-dose melphalan autologous stem cell transplant.

significant change in the diversity of the repertoire across time points or in relation to MRD or transplant status. A circle packing diagram to visualize the top five most abundant T-cell clones per sample in relation to phenotype, TCR αβ clonotype, and TCR β specificity group defined by a shared sequence motif within the complementarity-determining region 3 (CDR3), illustrates that T cells with the same clonotype exhibit varying phenotypes (Fig. 3d). It was also evident from this analysis that the most abundant T-cell clones from patients with sustained MRD negativity belonged to fewer specificity groups defined by their CDR3β amino acid sequence motif using GLIPH (grouping of lymphocyte interactions by paratope hotspots)[15] than patients with non-sustained MRD negativity indicating greater diversity to antigen specificity.

### Immune profiling of BMMCs by CyTOF mass cytometry
Using an orthogonal approach to single-cell sequencing, we leveraged CyTOF mass cytometry to profile the bone marrow microenvironment before and after exposure to lenalidomide maintenance. We analyzed a total of 503,663 individual cells (median, 7411 cells/sample; range, 272–117,823 cells/sample) and classified 22 different cell types (Fig. 4a and Supplementary Fig. 15).

Confirming what we observed using scRNA in the peripheral blood, CD8+ effector T cells were the most frequent T-cell subset at baseline and follow-up for patients with sustained MRD negativity and prior HDM-ASCT. Likewise, CD4+ central memory and effector memory T cells were the most abundant T-cell types on average in patients with non-sustained MRD negativity and no prior transplant

(Fig. 4b). Central memory T cells proliferate extensively and are predominantly located in the blood and secondary lymphoid tissues whereas effector memory T cells are less proliferative and are more commonly found in the spleen[16]. Unlike the blood, the overall proportion of T cells were higher in bone marrow from patients with non-sustained MRD negativity and no prior transplant and sustained MRD negativity and prior transplant. CD27− non-memory B cells were the dominant cell type at baseline in patients achieving sustained MRD negativity who had a prior transplant, while this finding was not seen in other subgroups (Fig. 4c). Finally, classical monocytes and conventional dendritic cells (DCs) were the most frequent myeloid cell type and were proportionally higher in patients without prior HDM-ASCT (Fig. 4d).

Differential abundance analysis comparing patients by their MRD-negative status and HDM-ASCT history at baseline and follow-up revealed differences not seen in the blood. While switched memory B cells were higher in patients with sustained MRD negativity at follow-up time points (Fig. 4e and Supplementary Fig. 16), CD27− non-memory B cells were more abundant in patients with prior HDM-ASCT at baseline (Fig. 4f). CD4+CD8+ T cells were more frequent in patients with non-sustained MRD negativity at baseline and in patients with prior HDM-ASCT at follow-up (Fig. 4e, f). Comparison of the percent change in frequency of cell types from baseline to follow-up demonstrated that only conventional DC (Wilcoxon rank-sum test $P = 0.022$) and NK T cells (Wilcoxon rank-sum test $P = 0.033$) significantly increased in patients with prior HDM-ASCT (Fig. 4g).

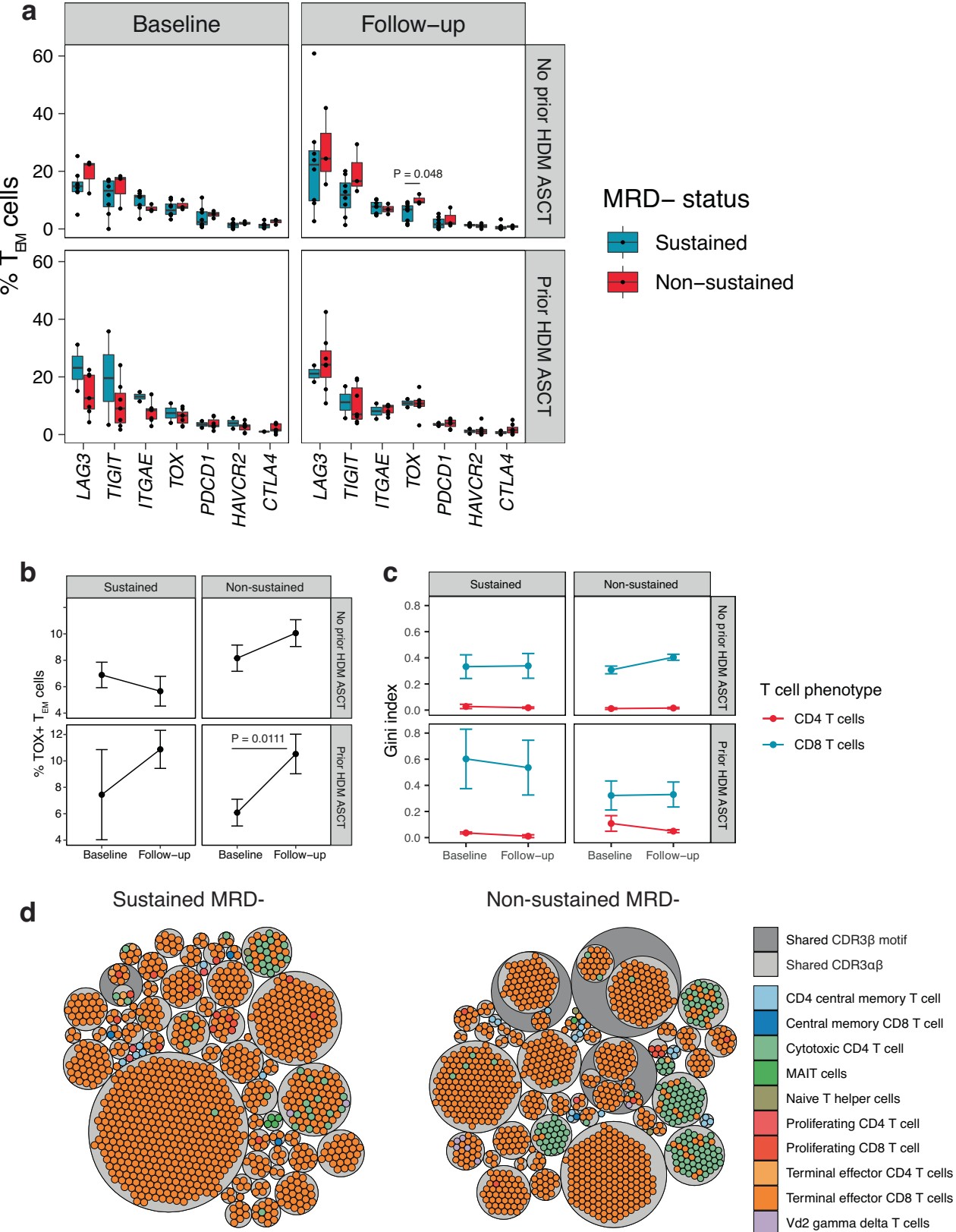

## T-cell receptor β repertoire analysis of the blood

TCR β sequencing was performed on 24 PBMC samples collected at two time points from 13 patients (9 non-sustained and 4 sustained MRD negative). A total of 1,311,455 unique, productive amino acid CDR3β sequences were identified (median, 41,504 sequences/sample; range, 15,360–133,160 sequences/sample). Repertoire diversity was assessed by the Gini index where 0 indicates the most diversity and 1 indicates the least. We observed significantly greater TCR β diversity within baseline samples among patients who had no prior history of HDM-ASCT (Wilcoxon rank-sum test $P = 0.003$) while diversity was similar across MRD groups (Wilcoxon rank-sum test $P = 0.940$, Fig. 5a). TCR β diversity did not significantly change following lenalidomide

**Fig. 3 | Analysis of circulating T cells expressing markers of exhaustion and T-cell receptor repertoire diversity. a** Abundance of terminal effector memory T cells (T_EM) expressing known exhaustion markers in cancer. **b** The mean percentage of TOX + T_EM cells and standard error at baseline and follow-up according to minimal residual disease (MRD) negative status and transplant history. **c** Single-cell T-cell receptor mean Gini index and standard error showing the diversity of CD4+ and CD8+ T cells at baseline and follow-up according to MRD-negative status and transplant history. A higher Gini index corresponds to greater receptor diversity. **d** Circle packing plot showing individual T-cell phenotype (colored

circles) in relation to clonotype (light-gray circles) and shared antigen specificity group (dark gray circles). Asterisks indicate a two-sided Wilcoxon rank-sum test $P < 0.05$. N = 20 sustained MRD- (16 no prior HDM-ASCT, 4 prior HDM-ASCT) and 20 non-sustained MRD- (6 no prior HDM-ASCT, 14 prior HDM-ASCT) patient samples. In the box plot, the ends of the whiskers represent 1.5 times the interquartile range, the center represents the median, and the bounds of the box represent the first and third quartiles. Source data are provided as a Supplementary Data file. HDM-ASCT high-dose melphalan autologous stem cell transplant.

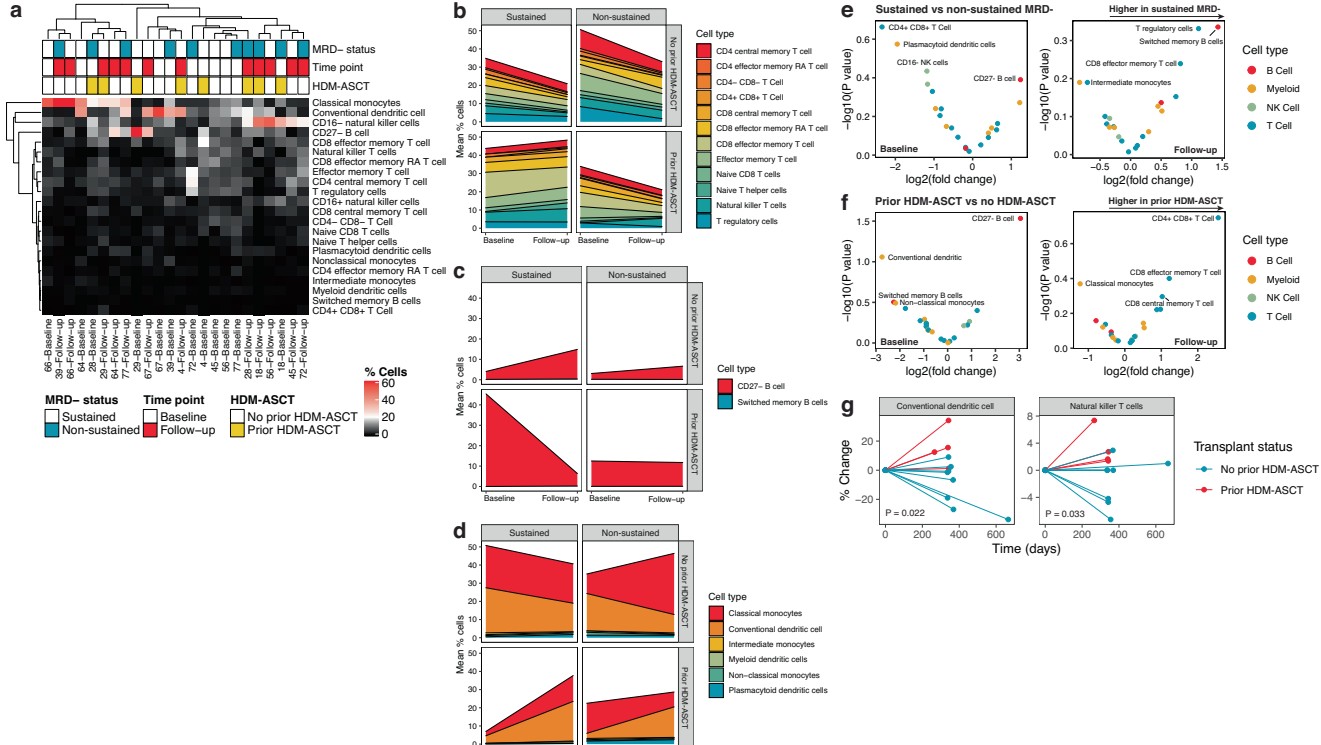

**Fig. 4 | CyTOF of bone marrow reveals differences in immune cell frequency according to minimal residual disease (MRD)-negative status and transplant history. a** Heatmap showing the frequency of immune cell subsets per patient. **b** Area plot showing the mean frequency of T cells, **c** B cells, and **d** myeloid cells at baseline and follow-up according to the patients' MRD-negative status and transplant history. **e** Volcano plot comparing differentially abundant bone marrow cells between patients achieving sustained and non-sustained MRD negativity at baseline (left) and follow-up (right). **f** Volcano plot comparing differentially abundant bone marrow cells between patients with prior history of transplant or not at

baseline (left) and follow-up (right). **g** Spider plot showing the percent change in cell frequency between baseline and follow-up time points according to the patients' transplant history. Only cell types that are significantly increased or decreased over time (Wilcoxon rank-sum test $P < 0.05$) are shown (there were no significant changes in cell frequency in relation to MRD-negative status). $N = 7$ sustained MRD- (5 no prior HDM-ASCT, 2 prior HDM-ASCT) and seven non-sustained MRD- (4 no prior HDM-ASCT, 3 prior HDM-ASCT) patient samples. Source data are provided as a Supplementary Data file. HDM-ASCT high-dose melphalan autologous stem cell transplant.

treatment in both the transplant (Fig. 5b, Wilcoxon rank-sum test $P = 0.072$) and MRD groups (Fig. 5b, Wilcoxon rank-sum test $P = 0.439$). To identify TCR β sequences specific to patients with sustained MRD negativity, we identified sequences unique to this subgroup that were not expected by chance using Fisher's exact test. This revealed a total of 74 unique, productive TCR β sequences specific to patients with sustained MRD negativity with frequencies ranging from $2 \times 10^{-4}$–$0.1 \times 10^{-1}$ (Fisher test $P < 0.05$, Fig. 5c). Surprisingly, these sequences were relatively common among patients with sustained MRD negativity, ranging from 28 to 44 sequences per patient. In contrast, only 23 TCR β sequences were found to be specific to patients with non-sustained MRD negativity (Fisher test $P < 0.05$). GLIPH amino acid sequence motif analysis revealed nine recurrent sequence motifs among the differentially abundant TCR β CDR3 sequences indicating increased likelihood of shared antigen specificity (Supplementary Data 6 and Supplementary Fig. 17).

To further investigate these 74 differentially detected sequences, we queried them in the VDJdb[17], McPAS-TCR[18], PIRD TBAdb[19], and LymphoSeqDB[20] databases to reveal known antigen specificities. This uncovered 14 CDR3β amino acid sequences with known antigen specificity to a variety of microbial antigens and one human antigen (Supplementary Data 6). We also searched for the CDR3β amino acid sequences within our scRNAseq dataset to determine if the receptor was associated with a particular T-cell phenotype. Due to the shallow depth of single-cell sequencing, this revealed only 4 CDR3β amino acid sequences, limiting our ability to draw further conclusions. To determine the prevalence of the CDR3β amino acid sequencing among 55 healthy individuals ages 0–90 years, we searched for the sequences in LymphoSeqDB database[20] and reported the frequency of healthy individuals who share this same sequence (Supplementary Data 6). Finally, we used OLGA to compute the generation probabilities (Supplementary Data 6)[21]. Together, these analyses revealed that T-cell

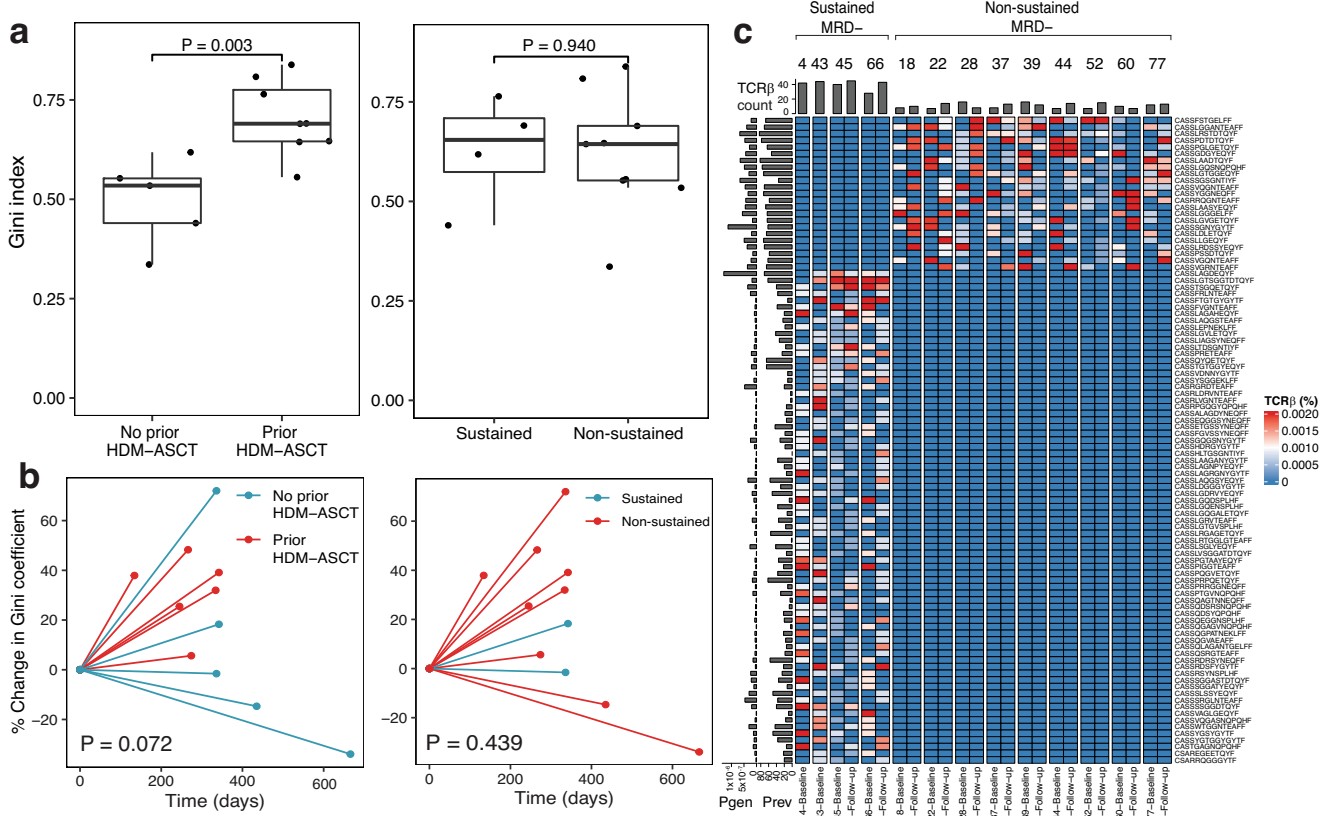

**Fig. 5 | T-cell receptor β (TCRβ) sequencing of blood reveals unique T-cell receptors to patients with sustained minimal residual disease (MRD) negativity. a** Box plots comparing the TCRβ diversity according to MRD and transplant status using the Gini index, where 1 equals the most clonal and 0 indicates the most diverse. **b** Spider plot showing the percent change in the Gini index across baseline and follow-up time points. **c** Heatmap showing the frequency of TCRβ sequences specific to patients with sustained MRD negativity (*n* = 74 sequences) or non-sustained MRD negativity (*n* = 23) that are not expected by chance (two-sided Fisher's exact test <0.05). Pgen, progability of generating CDR3β sequence (Olga);

Prev, prevalence of CDR3β sequence in healthy population (LymphoSeqDB). Two-sided Wilcoxon rank-sum test *P* values are shown. *N* = 6 sustained MRD- (4 no prior HDM-ASCT, 2 prior HDM-ASCT) and 18 non-sustained MRD- (6 no prior HDM-ASCT, 12 prior HDM-ASCT) patient samples. In the box plot, the ends of the whiskers represent 1.5 times the interquartile range, the center represents the median, and the bounds of the box represent the first and third quartiles. Source data are provided as a Supplementary Data file. HDM-ASCT high-dose melphalan autologous stem cell transplant.

receptor β sequences that are unique to patients with sustained MRD negativity are infrequently found in healthy individuals and have a relatively low CDR3 sequence generation probability. This suggests that these sequences are unlikely to be shared because they are prevalent in the human population or have a high probability of sequence generation.

**Comparison with healthy and precursor MM BMMC**

To understand how the immune microenvironment changes in relation to MM disease progression, we compared our scRNAseq datasets with a previously published scRNAseq dataset from ref. 14, examining BMMC from 9 healthy donors, 5 patients with MGUS, 11 patients with SMM, and 7 patients with untreated MM[14]. Although two different tissue types were compared in this analysis, we found good correlation between the frequency of BMMC and PBMC when measured from the same individual on the same day (*r* = 0.54, Supplementary Fig. 18). The similarity between BMMC and PBMC may suggest some degree of hemodilution, obscuring the differences in the immune cell frequencies between the two tissue types. After applying the same bioinformatics workflow and correcting for batch effects, we were able to directly track the frequency of immune cell subsets from healthy individuals, precursor MM, and active disease (Fig. 6a and Supplementary Fig. 19). We also compared our CyTOF BMMC dataset to the published scRNAseq BMMC dataset to reveal similar findings (Supplementary Fig. 20). Hierarchical clustering of cell counts averaged

across cell types clustered patients by precursor and active disease states as well as transplant history (Fig. 6b). Pairwise Pearson correlation of immune cell frequency revealed a positive correlation between healthy bone marrow and all subgroups, but the strongest correlation was between healthy donors and patients achieving sustained MRD negativity and no prior HDM-ASCT (*r* = 0.85, Fig. 6c, d and Supplementary Fig. 21).

## Discussion

In patients diagnosed with MM, disease control is driven by a complex interplay between tumor cells, the surrounding immune microenvironment, and ongoing therapy. However, differences in the immune microenvironment between short-term and long-term responders is not fully understood. Although HDM-ASCT has consistently been shown to prolong PFS[22–24], it causes severe and prolonged immunosuppression that could impact the ability of the immune system to effectively control the disease[25]. In this study, we comprehensively profiled the immune microenvironment of patients with newly diagnosed MM receiving lenalidomide maintenance and compared patients achieving MRD negativity at 1 year post treatment to those unable to maintain or who never achieved an MRD-negative state. To account for the heterogeneity in treatment histories, we examined patients with and without HDM-ASCT separately.

We observed differences in the composition of the immune cells in both the blood and bone marrow between patients with sustained

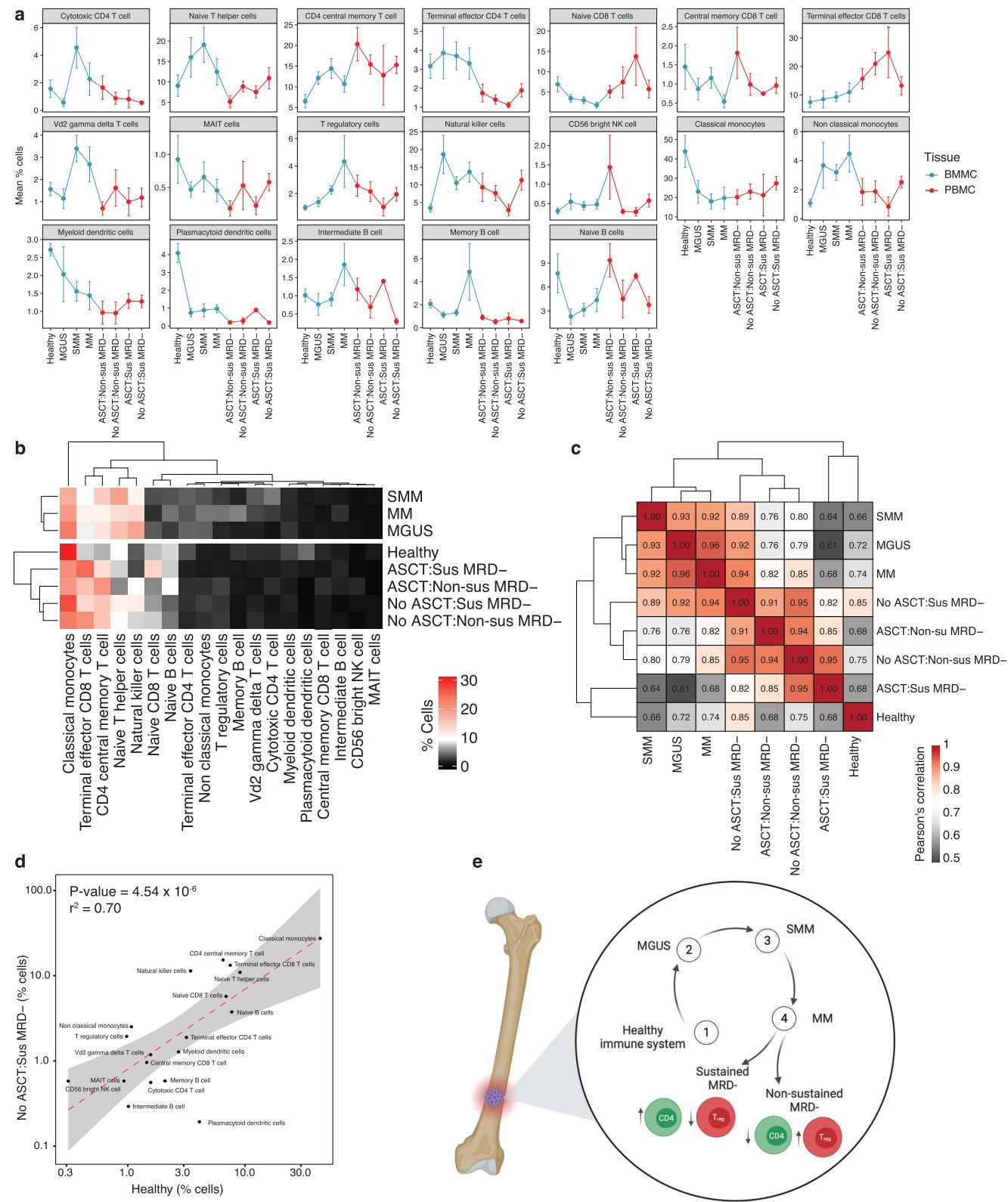

versus non-sustained MRD negativity that varied according to history of HDM-ASCT. For example, patients without a history of transplant and sustained MRD negativity had an increase in circulating CD4+ T cells, higher CD4/CD8 ratio and reduced number TOX + $T_{EM}$ cells. In contrast, patients with a transplant history and sustained MRD negativity tended to have more circulating CD8+ T cells and lower CD4/CD8 ratio. These differences continued to persist a year post-transplant. A reduced CD4/CD8 ratio has also been previously reported in the blood

and bone marrow in patients with MM[26] as well as in the bone marrow of patients undergoing an autologous stem cell transplant[27]. Increased T-cell expression of *TOX*, a crucial transcription factor involved in T-cell exhaustion, has recently been reported in both the blood and marrow of patients with MM compared to healthy controls[28]. However, the changes due to HDM-ASCT preclude any consistent finding regarding the correlation of T-cell phenotype with sustained MRD response within the current analysis.

**Fig. 6 | Comparison of immune cell frequency in the bone marrow and blood among healthy donors as well as patients with various stages of myeloma measured by scRNAseq. a** Line plot showing the mean and standard error of the cell frequency by cell type (plasmablasts excluded) in healthy bone marrow, precursor myeloma, untreated myeloma as well as blood from patients after receiving lenalidomide maintenance. **b** Heatmap and unsupervised hierarchical of cell frequency for each cell type averaged across subgroup. Samples are grouped according to K means clustering ($k = 2$). **c** Pairwise Pearson's correlation matrix comparing the mean cell frequency of each of the measured immune cell subsets. Higher numbers (red) indicate greater similarity of samples derived from patients with the shown diagnosis. **d** Scatter plot and linear regression analysis showing correlation between immune cell frequency in healthy BMMC versus myeloma

without prior HDM-ASCT achieving sustained MRD negativity. Line of best fit is shown in red and 95% confidence interval is indicated in gray. The linear regression $P$ value and coefficient of determination are shown. **e** Summary of major differences observed between sustained and non-sustained MRD-negative patients (created with BioRender.com). $N = 9$ healthy donors, 5 patients with MGUS, 11 patients with SMM, and 7 patients with untreated MM, 20 patients with sustained MRD- (16 no prior HDM-ASCT, 4 prior HDM-ASCT), and 20 patients with non-sustained MRD- (6 no prior HDM-ASCT, 14 prior HDM-ASCT). Source data are provided as a Supplementary Data file. Non-sus non-sustained MRD negative, Sus sustained MRD negative, ASCT autologous stem cell transplant, BMMC bone marrow mononuclear cells, PBMC peripheral blood mononuclear cells.

We also observed differences in sustained versus non-sustained MRD negativity that did not appear to be influenced by HDM-ASCT. For example, the frequency of circulating Treg cells significantly declined after a year of maintenance therapy in patients with sustained MRD negativity, regardless of their transplant status. Although there is some debate as to whether Treg cells are increased in patients with MM, there is evidence to suggest that they have an inhibitory effect on the proliferation and activation of effector T cells[29,30]. Other studies have shown that lenalidomide decreases the frequency of Treg cells compared to patients receiving a proteasome inhibitor and that the degree of reduction correlates with depth of response[31]. In this study, it appears that while Treg cells are of similar frequency in patients with sustained and non-sustained MRD negativity, but are more likely to decline in durable responders after treatment with lenalidomide.

In terms of the TCR β repertoire of the peripheral blood, its diversity did not appear to have a substantial impact on the duration of response. Although no history of HDM-ASCT was associated with increased diversity before the onset of maintenance, there was no association with MRD status. Furthermore, there did not appear to be a significant change in the diversity after lenalidomide exposure in either group. Instead, we detected highly specific TCR β sequences that were unique to the repertoire of patients with sustained MRD negativity. Of the 74 sequences identified, 38–59% were present in each patient. Further investigation would be necessary to determine the antigen specificity of these TCR β sequences.

Our results support prior observations showing that the percentage of classical monocytes is reduced in patients with MM compared to healthy controls[32]. This appears to be independent of the history of HDM-ASCT, as classical monocytes are reduced even in patients who have not undergone a transplant. In contrast, non-classical monocytes are increased in patients with MM, which has also been reported previously[33,34]. It has been hypothesized that non-classical monocytes may produce cytokines such as IL-6 that may enhance myeloma cell growth[35]. Zavidij et al. found evidence to suggest that bone marrow CD14+ monocytes exhibit compromised MHC class II surface expression and can suppress T-cell activation leading to accelerated proliferation of MM cells[14].

Finally, we observed a strong, positive correlation between immune cell frequencies through a comparison of healthy donor bone marrow and PBMCs from patients who had achieved sustained MRD negativity without a history of HDM-ASCT. This correlation was weaker for patients with a prior history of HDM-ASCT. Patients with untreated MM showed the lowest degree of similarity to healthy donors. This finding suggests that the immune system of patients achieving sustained MRD negativity could ultimately recover to a healthy state, even in the setting of ongoing treatment with lenalidomide (Fig. 6e). It also suggests that the immunosuppressive impact of HDM-ASCT is prolonged up to 1 year that may represent a potential advantage for the onset and accelerated expansion of more aggressive subclone[36].

All samples and clinical data were collected as part of a large prospective, longitudinal single-arm maintenance clinical trial. This ensured high-quality data through consistent procedures for

monitoring and sample capture, as well as pre-planned assays. Because the treatment landscape for newly diagnosed MM is emerging rapidly and a range of treatment options are considered standard-of-care, the study protocol allowed patients to receive "dealer's choice" combination therapy (with or without HDM-ASCT) prior to enrolling in this maintenance trial. The intent was to enable us to study patients with "real world" outcomes. To account for heterogeneity in up-front therapies, we studied patients with and without HDM-ASCT separately, as this treatment is known to have a significant, long-term impact on the immune microenvironment. Since this inherently resulted in the comparison of a smaller number of patients, additional studies in a larger population are needed to confirm and expand on our findings. Furthermore, our observations do not inform on the possible function of lenalidomide, or whether these immune correlates are a consequence of low disease burden, allowing a return to a healthier immune system, or whether they indicate a contribution from the immune microenvironment to disease control. Based on our clinical experience, we believe all these factors play a role; however, further investigations are needed to better define the exact molecular mechanisms. Finally, it is important to recognize that while our definition of sustained MRD negativity clearly stratified patients according to their PFS, a more sensitive measure of MRD testing using next-generation sequencing may have classified fewer patients as achieving sustained MRD negativity in our study.

In summary, the host immune system is critical for the eradication of tumor cells both directly and through its interaction with immunotherapy. Through comprehensive immune profiling of the bone marrow and blood from patients with newly diagnosed MM receiving lenalidomide as maintenance, we uncovered distinct differences in immune cell composition specific to patients with sustained MRD negativity that vary according to exposure to high-dose melphalan. Additional functional studies are needed to validate these associations to determine cause and effect. We also reveal similarities of healthy bone marrow to the blood of patients with sustained MRD negativity in contrast to patients with untreated MM. Our findings support the hypothesis that the immune microenvironment influences the duration of treatment response and suggests that the immune system of patients achieving a durable remission may recover to a premalignant state.

## Methods
### Study population
This study was approved by the Memorial Sloan Kettering Cancer Center Institutional Review Board. Informed consent was obtained from all participants in accordance with the Declaration of Helsinki Protocol. Study participants were not compensated for their participation. Immune profiling was performed on 23 patients with newly diagnosed MM who participated in the single-arm, phase II clinical trial evaluating the effect of lenalidomide as maintenance therapy (Fig. 1a, NCT02538198)[5]. All clinical trial participants received lenalidomide 10 mg days 1–21 every 28-day cycle. MRD testing was performed at enrollment and annually until the progression or end of the study on

first-pull bone marrow aspirates using 10-color, single-tube flow cytometry or 8-color, two-tube flow cytometry in accordance with the EuroFlow Consortium (sensitivity ≥ $10^{-5}$)[37]. In most cases, blood and bone marrow samples were collected on day 1 of a treatment cycle. Since the aim of our study was to identify immune features that correlate with durable treatment response, we compared 12 patients with sustained MRD negativity to 11 patients who lost or never achieved MRD negativity (referred to as non-sustained MRD negativity). Sustained MRD negativity was defined as having at least two MRD-negative results measured 1 year apart. For patients with non-sustained MRD negativity, we selected patients who had progressed within 2 years of initiating lenalidomide maintenance. Progressive disease was defined by the international uniform response criteria for MM[38].

### Sample processing
Sixty bone marrow aspirates from 17 patients and 40 blood samples from 20 patients were analyzed. BMMCs and PBMCs were isolated by Ficoll-Paque gradient centrifugation, suspended in 90% fetal bovine serum (FBS) and 10% dimethyl sulfoxide, and cryopreserved in liquid nitrogen until analysis. Bone marrow plasma was obtained by centrifugation and stored at −80 °C. Each immunoassay was performed at two time points, once before the start of lenalidomide maintenance (median 7 days before) and once again during treatment (median 342 days after, Fig. 1b). Immune profiling using all three technologies was not possible for every patient due to limited sample availability. One assay type was performed on 8 patients, two assay types were performed on 6 patients, and all 3 assay types were performed on 10 patients.

### Single-cell RNA sequencing
Single-cell gene expression profiling was performed on 40 PBMC samples collected from 20 patients. PBMCs were thawed, washed with warm RPMI supplemented with 10% FBS, and counted using a hemocytometer and Countess II Automated Cell Counter (Applied Biosystems). The final cell count was determined by taking the average total cell count between the manual and automated methods. The mean cell viability by trypan blue stain for all samples was 76%. Cell suspensions were loaded into a Chromium Controller (10X Genomics) to achieve a target cell recovery of $10^3$ cells per sample. Upon single-cell capture, libraries were prepared using the Chromium Single Cell 5' Gene Expression and V(D)J Enrichment Kit for Human T and B Cells v1.0 (10X Genomics). Barcoded libraries were pooled, and 100 bp paired-end sequencing was performed on a NovaSeq 6000 (Illumina).

Sequencing files were demultiplexed, aligned to the human reference genome GRCh38, and feature barcode matrices were generated using cell Ranger v4.0.0 (10X Genomics). Cells with less than 500 unique molecular identifiers (UMI), less than 250 aligned genes, or greater than 20% mitochondrial genes detected were excluded. UMI count normalization, feature selection, and scaling was performed using the SCTransform function from the Seurat v4.0 package (Supplementary Methods)[39]. Dimensionality reduction using UMAP and trajectory analysis was performed using monocle3[40]. Reference-based mapping implemented by Azimuth from the Seurat package was used to perform automated cell classification of immune cell subtypes using a human PBMC reference dataset (Supplementary Data 7)[39]. Due to insufficient power for statistical comparisons, rare cell types were excluded if represented by less than 200 cells in the entire dataset (0.2% of cells analyzed). Integration with external scRNAseq datasets (dbGaP Study Accession: phs001323.v2.p1, Supplementary Data 8)[14] was performed using the FindIntegrationAnchors function with dimensional reduction method set to reciprocal PCA from the Seurat v4.0 package. Differential gene expression analysis was carried out between sustained and non-sustained MRD-negative patients at baseline and follow-up time points for each cell subset using the EdgeR-LRT method implemented by the Libra v1.0 R package[41].

The diversity of the paired TCR αβ repertoire was estimated using the Gini index, which ranges from 0 to 1 with a higher number corresponding to a more clonal repertoire[42]. GLIPH was used to identify antigen specificity groups from shared CDR3 amino acid sequence motifs within the TCR β chain[15]. Visualization of the T-cell phenotype in relation to shared TCR αβ and GLIPH specificities was accomplished using the TCellPack R package (https://github.com/davidcoffey/TCellPack).

### CyTOF mass cytometry
CyTOF mass cytometry was used to analyze 26 BMMC samples from 13 patients. BMMC were thawed, washed, and barcoded according to manufacturer recommendations (Fluidigm). Barcoded cells were stained using a custom 19-marker panel designed to characterize innate and adaptative immune cell subtypes (Supplementary Table 1 and Supplementary Data 9). Stained cells were acquired on a CyTOF Helios mass cytometer (Fluidigm).

Automated cell classification was performed using the Astrolabe Mass Cytometry Platform (Astrolabe Diagnostics, Inc.). Briefly, immune subsets were clustered into self-organizing maps using the FlowSOM v2.8 package[43] and labeled using the Ek'Balam algorithm[44]. Cell subset definitions follow previously reported definitions of the healthy human immune system and the Human ImmunoPhenotyping Consortium (Supplementary Data 10)[45,46]. Cell types were excluded if there were less than three cells in at least half of all samples.

### TCR β sequencing of PBMCs
Genomic DNA was extracted from 24 PBMC samples using the AllPrep DNA/RNA kit (Qiagen, Cat. No. 80204) and quantified by Nanodrop (Thermofisher Scientific). Library preparation was performed using the hsTCRB Immunoseq kit (Adaptive Biotechnology) and paired-end sequencing was performed on a NextSeq 500 (Illumina). TCR β chain V, D, and J gene alignment and calculation of complementarity-determining regions 3 (CDR3) frequency was performed using the ImmunoSEQ Analyzer (Adaptive Biotechnology). Repertoire diversity was computed using the Gini index using the LymphoSeq v1.14.1 package (Supplementary Data 11)[20]. Public TCR β sequences we queried in the VDJdb[17], McPAS-TCR[18], PIRD TBAdb[19], and LymphoSeqDB[20] databases using the immunarch v0.6.9 package (https://immunarch.com/).

### Statistical methods
Survival curves were estimated using the Kaplan–Meier method, and the difference between subgroups was evaluated by the long-rank test. Differential abundance analysis was performed using a negative binomial generalized log-linear model implemented by the EdgeR v3.40.2 package[47]. Single-cell differential gene expression was performed using EdgeR-LRT pseudobulk method implemented by the Libra v1.0 R package[41]. Wilcoxon rank-sum test was performed to compare the change in immune cell frequency across patient groups with two-sided $\alpha = 0.05$. P values were adjusted for multiple comparisons using the Benjamini–Hochberg method. Correlation between immune cell frequency from separate datasets was performed using the Pearson correlation coefficient and linear regression analysis. Fisher's exact test was used to determine the probability of finding TCR β sequences specific to patients with sustained MRD negativity. All analyses were performed in R v4.1.

### Reporting summary
Further information on research design is available in the Nature Portfolio Reporting Summary linked to this article.

## Data availability
Processed scRNAseq has been made publicly available through CReSCENT: CanceR Single Cell ExpressioN Toolkit (https://crescent.

cloud, CRES-P31), a web portal for standardized analysis and exploration of scRNAseq data from cancer studies[48]. Single-cell gene expression matrix files have also been deposited in Figshare (https://doi.org/10.6084/m9.figshare.23816625.v1). The raw scRNAseq data are protected and are not available due to data privacy laws. scRNAseq and CyTOF cell counts are available in the Supplementary Data provided with this paper. Single-cell V(D)J calls are available in the Supplementary Data file. TCR β sequencing is available within the ImmuneAccess database (https://clients.adaptivebiotech.com/immuneaccess, https://doi.org/10.21417/DGC2023NC). Previously published scRNAseq from the bone marrow of healthy donors and patients MGUS, SMM, and MM were accessed from the Gene Expression Omnibus under accession code GSE124310[14]. Quiered T-cell receptor databases are accessible through the R packages LymphoSeqDB (https://bioconductor.org/packages/LymphoSeqDB) and immunarch (https://immunarch.com/). Source data are provided with this paper.

## Code availability

R code for analyzing scRNAseq combined with V(D)J sequencing, CyTOF mass cytometry, and TCRβ sequencing is available on GitHub (https://github.com/UM-Myeloma-Genomics/Immunophenotypic-correlates-of-sustained-MRD-negativity)[49]. Visualization of single-cell T-cell phenotypes in relation to shared TCR clonotype and CDR3β motifs was performed using TcellPack v2.2, a custom R package (https://github.com/davidcoffey/TCellPack)[50].

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

## Acknowledgements

This research was funded by the Immune Network of Excellence grant from the Multiple Myeloma Research Foundation (A.L., D.K., T.P., D.G., S.G., and O.L.), the Sylvester Comprehensive Cancer Center NCI Core Grant (P30 CA 240139), the Memorial Sloan Kettering Cancer Center NCI Core Grant (P30 CA 008748), Fred Hutch NCI Core Grant (P30 CA 015704), the Riney Family Multiple Myeloma Research Program Fund, and Intramural Research Programs of the FDA/OCE and NIH/NCI. The clinical trial component was supported by Celgene/BMS. The Fred Hutch Genomics Shared Resource assisted with next-generation sequencing. The Scientific Computing Infrastructure at Fred Hutch was funded by an ORIP grant S10OD028685.

## Author contributions

A.L., D.K., T.P., D.G., S.G., O.L., and D.C. conceived and designed the study. D.C., Y.X., T.D., B.L., A.R., S.K., A.C., H.X., E.S., H.C., B.D., and H.W. collected the data for the study. D.C., F.M., E.G., J.D., P.L., and Y.Z. contributed to data analysis. D.C. wrote the paper with input from all authors. The final version of the manuscript was reviewed by all authors.

## Competing interests

O.L. acknowledges funding from: NCI/NIH, FDA, LLS, Rising Tide Foundation, Memorial Sloan Kettering Cancer Center, MMRF, IMF, Paula and Rodger Riney Foundation, Perelman Family Foundation, Amgen, Celgene, Janssen, Takeda, Glenmark, Seattle Genetics, Karyopharm; has received honoraria for scientific talks/participated in advisory boards for Adaptive, Amgen, Binding Site, BMS, Celgene, Cellectis, Glenmark, Janssen, Juno, Pfizer; and served on Independent Data Monitoring Committees (IDMC) for international randomized trials by: Takeda, Merck, Janssen, Theradex. S.G. acknowledges funding from U24 CA224319, U01 DK124165, and from the Mount Sinai Tisch Cancer Institute Cancer Center NCI Core Grant P30 CA196521; and reports consultancy and/or advisory roles for Merck and OncoMed; and research funding from Bristol-Myers Squibb, Genentech, Boehringer-Ingelheim, Celgene, Janssen R&D, Pfizer, Takeda, and Regeneron. The remaining authors declare no competing interests.
