## [Peer Review File · Nature Communications]

Immunophenotypic correlates of sustained MRD negativity in patients with multiple myelomaREVIEWER COMMENTS

Reviewer #1 (Remarks to the Author):

In the study by Coffey et al., the authors analyzed samples from 23 patients who went on a clinical trial for maintenance with lenalidomide. They longitudinally characterized the immune landscape before and one year after lenalidomide exposure.

I commend the authors for their admirable work by combining single-cell RNA and TCR sequencing of the PBMCs with CyTOF on the BMMCs to study the immune landscape of these patient samples.

I have some comments about the study design, which I believe prevents definite conclusions given the heterogeneity of the induction regimens received by the patients and how this could affect the depth of response and MRD status.

- First, these samples are from PBMCs of the MM patients, which are mainly depictive of systemic immune repertoire, and we don't know how similar or different they are from the BMMCs of these patients. I understand that the authors performed CyTOF on samples from 13 patients and propose they are similar. However, it is a known issue for the bone marrow samples that they could be hemodiluted in such settings. Could the authors comment and address this, as I haven't seen a sufficient mention of this point?

- Second, I have a problem with the authors' definition of sustained vs. unstained MRD-ve. If patients with unsustained MRD-ve never been -ve from the beginning, then why were they included? They do not meet the criteria of MRD -ve state. This creates a misunderstanding of the results. I advise changing the term to something like successful vs. failed MRD state or any other accurate definition as sustained implies that a measure was already achieved and continues to occur, which is not the case here.

- In Table 1, the authors performed a chi-square test on categorical values, which is not the most appropriate test to be used but instead fisher exact given the small sample and frequency of events. The authors should use fisher's exact test to get an accurate p-value. The authors should also include an analysis of the difference between patients who received IMids and those who did not. Same for carfilzomib, which is known to be a more potent PI than bortezomib.

- In the unsustained MRD negative group, 10 patients had persistent MRD positivity from the start of maintenance and 2 patients who converted from MRD negative to positive within the first year of maintenance. However, in supplemental table 1. This data does not match with that in the main text.

- In Figure 1 A, 2 patients became MRD +ve in the sustained MRD group; why were they not included in the unsustained group? How was the cutoff determined, and why? How did these patients do in terms of disease progression?

- I see the sensitivity of the MRD testing was 10^{-5} , which is less sensitive than the current threshold. The authors need to acknowledge this limitation in the discussion and explain its potential relevance as it could have identified more MRD-positive patients.

- On page 8, line 213: "Likewise, CD4 central memory and effector memory T cells were the most abundant T cell types on average in patients with unsustained MRD negativity and no prior transplant (Fig. 4B)" While in page 6, the authors mention: "In contrast, a higher CD4:CD8 ratio was observed for all other comparisons, especially in patients with sustained MRD negativity without prior HDM-ASCT where CD4 central memory and naïve CD4 T cells were the most abundant lymphocyte." Could the authors comment on this part, as it looks contradictory, and discuss the significance and role of the central memory CD4 cells in the immune microenvironment?

- On page 9, line 232, the title is "T cell receptor β repertoire of the bone marrow and blood." I don't see anything on bone marrow in this paragraph. Why is that?

- The authors discussed the correlation between their analysis of PBMCs and external cohorts of BMMCs in normal and MGUS patients. I see that the comparison is inherently limited given that these are two different tissues and the impact of a batch effect in such analysis. The latter could be accounted for, but the former is hard to account for in such a setting, and I don't know how reliable this comparison could be in deriving any conclusions. Moreover, the authors mentioned that normal and MGUS samples were more correlated to patients without a transplant in this study. What was the statistical test done, and was it significant to lead to this conclusion?

- The authors did not mention in their discussion the finding that in both the bone marrow and blood, there is an abundance of classical and non-classical monocytes, which are shown in multiple studies to be immunosuppressive, in patients with no prior transplant. The authors should also comment and discuss this and review the current literature on this topic to present the complete picture.

- I feel that the authors are leading their conclusions to the point that HSCT is not as effective and has a deleterious effect on the immune microenvironment. The authors need to mention the fact that randomized clinical trials like the IFM and Forte have shown that HSCT is still providing better outcomes in terms of PFS, the same surrogate the authors used here. It is essential to provide these results with what they also propose.

- Finally, the authors should have complemented this with genomic and transcriptomic data that they probably have on the CD138+ tumor cells or from FISH data. It will give a more comprehensive picture rather than this divided approach on the predictors of MRD status. I believe this is an essential part to be added to the data.

Reviewer #2 (Remarks to the Author):

In this manuscript Coffey et al. profiled the immune system of patients with newly diagnosed myeloma receiving lenalidomide maintenance with the main objective to correlate long term treatment response with specific immune changes. By performing CyTOF, single cell RNA sequencing and TCR analysis in longitudinal collected samples before and 1 year after the initial treatment the authors observed that the composition of the T cell population in the bone marrow of these patients somehow varies in patients achieving sustained minimal residual disease (MRD) negativity versus patients who never achieved or were unable to maintain MRD negativity.

The authors found a higher increase in terminal effector CD8+ T cells resulting in a reduced CD4+:CD8+ ratio in the marrow of patients maintaining long term MRD negativity. While the study is somehow interesting, I believe that the scientific conclusions cannot be fully supported from the presented results. It is extremely well known that lenalidomide mechanisms of action is to increase cytotoxic T cell expansion and that once MM cells are eradicated the BM immune- compositions better resembles the one found in cancer free donors. This is also somehow represented from the fact that the authors reported that CD27- non-memory B cells was found the dominant cell type at baseline in patients achieving sustained MRD negativity who had a prior transplant, while this finding was not seen in other subgroups further. From the presented data it seems like that a healthier BM composition maybe essential to maintain MRD negativity upon Lenalidomide treatment. In my point of view the authors should further focus on understanding how the initial response to lenalidomide (early cycles) of the immune environment can correlate with longer MRD negativity. This approach could be extremely informative to understand the ability of patients to deeply respond to the treatment based on the immune changes more than on the plasma cell clone diversity.

Specifically:

- 1) Did the authors collect peripheral blood or bone marrow samples at earlier time points to assess whether initial response to lenalidomide treatment associated with terminal effector CD8+ T cell expansion, NKT can correlate with prolonged MRD negativity? In my point of view the analysis of early time points could be more informative about lenalidomide mechanisms of action and the importance of the microenvironment response to keep the tumoral cell under control, independently of their initial genetic profile. Unfortunately, after 1 year of treatment it is not clear if the presence/absence of CD8+T cells and changes in the other immune subsets is the cause or only the consequence of long MRD negativity. It is also not clear of how this information can help understanding the role of the environment to keep disease under control.
- 2) Is the composition of the immune environment changes once patients actively progress?
- 3) The legends of all the supplementary figures are not clear and it is extremely hard to follow what the authors want to show in the associated Figures.

Overall, this is an interesting manuscript exploring the importance of the immune environment in response to Lenalidomide maintenance therapy, but the novelty is somehow compromised from the limited time points analyzed.

Reviewer #3 (Remarks to the Author):

In this manuscript Coffey et al present data about the impact of immune microenvironment on MRD negativity achievement. The project is novel, but I have some major concerns:

Novelty is based on the methodology employed, but there is some defects in the design to get their objective. The results do not support completely the conclusions

- 1.- The MM population studied is very heterogenous and the treatments employed are assorted. This could make the conclusion weaker.
- 2.- The number of patients analyzed is small to get strong conclusions 12 and 11
- 3.- Probably the hypothesis should be reformulated to compare patients that relapse early and those that relapsed late. Because some patients achieving MRD negativity relapse
- 4.- Should be of interests to study a group of patients with MM in CR but without treatment with lenalidomide to define the effect of lenalidomide in the microenvironment

Reviewer #4 (Remarks to the Author):

Summary

This is a well written paper addressing an interesting question around the contribution of immune function to quality of disease response in treated myeloma patients, and is based on the analysis of a nice and novel dataset. The authors have undertaken single cell analysis of bone marrow and blood cells from patients with MM undergoing lenalidomide maintenance following initial therapy, employing a variety of tools: single cell RNA sequencing, V(D)J sequencing and TCRbeta sequencing of peripheral blood mononuclear cells and CyTOF analysis of bone marrow mononuclear cells. Each patient was sampled at start of maintenance, and then subsequently after a year of maintenance. The aim of the study was to uncover immunological features associated with sustained MRD negative response in patients receiving Lenalidomide maintenance. 12 patients had sustained MRD negativity, while 11 were either MRD positive throughout, or converted from MRD negative to positive within the year. Patients had received a variety of previous therapies but the strongest influence on their immune profiles was the receipt of high dose therapy and stem cell transplantation (SCT), that resulted in distinctive T cell features which persisted a year after maintenance. The authors report higher frequencies of naïve and central memory CD4 cells in patients without a prior history of transplant who had sustained MRD negativity, however no statistical information is given.

Major issues

Patient groups.

As acknowledged by the authors, the 2 groups of patients were unbalanced in several aspects: those with sustained MRD negativity were slightly younger, were less likely to have high risk features such as high risk genetics and ISS3, and less likely to have received SCT. They were also more likely to have received carfilzomib as part of their induction regimen (8/12) compared with the MRD positive group, of whom only 1 received a carfilzomib regimen as first line, while another received it at 4th line before proceeding to SCT. Thus the achievement of MRD negative response in the first place was influenced by disease features and treatment received, and the ability to sustain this quality of response with lenalidomide maintenance may have less to do with immune 'fitness' than with prior treatment history. In order to answer the question that the authors posed, ie. the immunological determinants of sustained MRD negative response, they would have to start with similar patient populations at a similar level of disease response, then compare those that sustained MRD negativity with those that didn't. Given the patient cohort that they have, however, the authors should at least acknowledge the possible confounding effect of the treatment imbalance on their results.

Results

Figure 1 is This is an excellent schematic outlining the cohort and patient datasets.

Single cell RNA sequencing data and Figure 2.

The authors have used a reference based mapping tool provided by Azimuth, and have omitted a gene expression clustering step in their scRNAseq analysis. By relying on the cell identities predicted by Azimuth the authors may miss groups of cells which may be enriched in specific conditions. For example, if a group of Treg cells are enriched in MRD+ versus MRD- patients, but all are uniformly given the label "Treg" by Azimuth, this observation would be lost. If however, Tregs were subjected to clustering analysis and these two groups of cells formed distinct clusters, this observation could be noted. Furthermore, the authors assume the phenotypic spectrum occupied by PBMC in their data is encompassed by Azimuth (which will only predict labels present in its reference), which may not be most appropriate. The author's decision to remove cell types present <200 times in the dataset (line 399) also seems needlessly restrictive.

There was also no differential expression analysis presented, which could easily occupy an entire figure by itself given the number of samples available for each group of patients. We would strongly recommend the Pseudobulk approaches outlined in Squair et al. (<https://doi.org/10.1038/s41467-021-25960-2>) for such analysis,

Figure 2A - B. It is unclear why trajectory analysis was performed, and it adds little to the manuscript.

Figures 2C - E and Fig S1 do not effectively show quantitative statistical differences between the patient groups at a per-patient level. Boxplots of proportions, with each point being a donor, would be more suitable.

It is unclear how genes were selected in Fig S2. Suggest display marker genes derived by differential expression.

Is the T cell clonality presented in Fig S3 correct? Why are naïve CD4 and CD8 T cells clonotypes, biologically non-clonal, shown as expanded? More information could be acquired by examining clonality beyond a binary >1 threshold. Defining deciles or larger expansion, for example as >10 cells, could help elucidate if any T cell subsets are particularly expanded.

Figure 3 and T cell exhaustion

There is insufficient information to substantiate the authors' claims to have identified exhausted T cells as outlined in Figure 2. The identification of exhaustion based on (co-)expression of single markers, each assessed in a binary fashion, can be misleading; the hallmark of an exhausted T cell is the prolonged co-expression of several co-inhibitory receptors, each of which may not be significantly expressed. The authors have identified 3 markers, but ICOS, LAG3 at least are also activation

markers. This is in keeping with the higher expression on proliferating T cells, again this is also seen in exhausted subsets and is simply an indication that the cells are responding to a stimulus.

Several key transcriptional markers of these cells, including PD1(PDCD1) or transcription factors associated with exhaustion like LAYN or TOX/TOX2 seem uninvolved or were not investigated. An exhaustion program is most likely to be detected in cell types with a phenotype resembling effector T cells, hence the absence of exhaustion markers in the Azimuth-predicted effector populations is noteworthy. Of those exhaustion-associated markers shown to be expressed, some of the highest expression being in MAIT and DN T cells (which are not canonically associated with the exhaustion pathway) might be a warning sign that these genes are being expressed absolutely at very low levels. The expression of a subset of exhaustion markers in proliferating T cells is promising, but the authors should note the higher sequencing depth achieved in proliferating cells will boost the number of detected genes and give an impression of higher per-gene expression (making the absence of PDCD1 in these cells even more noteworthy). Further, these markers may be expressed in proliferating CD4 because this includes proliferating Tregs.

Fig 3B should be re-done as a heatmap including classical transcription factors associated with exhaustion such as Tbet, TOX and LAYN.

Fig 3C: there is no definition as to what discriminates cells which are or are not exhausted (ie. a threshold or score)

TCR sequencing analysis

The analysis of TCR diversity is problematic unless the authors undertook subsampling to a fixed number of UMI defined TCR mRNA sequences? This is required in order to account for the wide variation in the actual number of TCR sequences obtained from each sample.

The packing diagrams (Fig 3E), which clearly show the phenotype of the same clonotype are nice, however, the continuous "heat" colour map is harder to read than a selection of diverse colors.

Publicity does not mean shared reactivity. Defining a shared TCRB as "shared TCR specific group" is not strictly accurate and the specificity of these clonotypes can only be speculated about at this point. If the authors wanted to pursue this route, could it be shown that such publicity in PB TCRB chain does not exist in age-matched healthy donors? If so, was it associated with a particular T cell phenotype in this dataset? Furthermore, it should be asked if any TCRs, particularly the TCRB chains shared between donors, were also found annotated in public TCR databases such as the VDJdb (<https://vdjdb.cdr3.net/>). Beyond a shared reactivity, TCR chains may be frequently identified because they possess a higher probability of generation (not every TCR chain has a similar likelihood of being generated during T cell development, with some being much more frequent than others). Perhaps the tool OLGA (<https://github.com/statbiophys/OLGA>) and associated manuscript could be employed here.

CytoTOF data, Figure 4

This is the only data to be obtained from BM samples, and the authors acknowledge that there are differences in subsets, but fail to match the cell populations in CyTOF those in the scRNA seq dataset. The CyTOF analysis would benefit from more exploratory clustering of the scRNAseq data. The interesting compositional findings from the CyTOF regarding NKT and cDC cells cannot be supported by the scRNAseq as these cell types were absent.

Fig 4B, C, D. Same comment as for Fig 2C, D, E. Can the authors show stats for the CD8 effector subset frequency in patients with prior SCT?

Fig 4B. The authors state that the overall proportion of T cells is higher in BM of patients with sustained MRD negativity and no SCT and in those with unsustained MRD negativity and prior SCT but surely it is the other way around?

In Fig 4E it looks as if Treg cells are higher in patients with sustained MRD negativity at follow up, whilst the single cell sequencing data in Fig 2H suggests that Treg cells actually reduce in these patients. Can the authors explain this apparent discrepancy in their findings?

Fig 5. The same caveats regarding public TCR beta chains described in Figure 3 apply to Fig 5C.

Figure 6. Comparison with healthy donors and precursor conditions.

The correlation between patients achieving MRD status and healthy donor marrows is interesting (although Fig 6B could require some per-donor statistics). As this is presented as one of the major findings of the manuscript, further analysis is required.

Fig 6C: Presume this plots median Pearson's correlation comparing disease group with healthy bone marrow? Given the wealth of sequencing data available, surely a more indepth analysis could yield information on whether if this is dominated by a subset of cell types which could serve as individual biomarkers vs whole bone marrow scRNAseq?

Machine learning approaches (feature selection) can determine which variables (here, cell types) contribute most to an outcome (here, correlation). Otherwise the value and relevance of this comparison is questionable.

Discussion

This largely re-iterates the descriptive narrative of the results section and provides no biological interpretation or insight into the clinical or otherwise significance of the findings.

The authors attempt a summary of the immune correlates of sustained MRD negativity in non- SCT and SCT patients, observing an increase in circulating CD4 cells, less exhausted T cells in non-SCT patients while in SCT patients had more circulating CD8 effectors and T cells with exhaustion phenotype. Aside from the lack of statistical data to back them up, these findings do not inform on the possible function of lenalidomide, or whether these immune correlates are as a result of the low disease burden, allowing a return to a more healthy immune system, or whether they indicate some sort of immune contribution to disease control.

The authors state there that "...history of HDT-ASCT was associated with a increased diversity before the onset of maintenance...", this is not what they report?

The authors need to differentiate, if they can, which immune features are indicative of sustaining an MRD negative response, from those that indicate an effect of lenalidomide therapy. The heterogenous nature of their patient cohort makes this problematic. Suggest that they focus on one cohort, preferably the non-ASCT cohort, where the immune landscape is not so skewed by the history of the ASCT.

REVIEWER COMMENTS

Reviewer #1 (Remarks to the Author):

In the study by Coffey et al., the authors analyzed samples from 23 patients who went on a clinical trial for maintenance with lenalidomide. They longitudinally characterized the immune landscape before and one year after lenalidomide exposure.

I commend the authors for their admirable work by combining single-cell RNA and TCR sequencing of the PBMCs with CyTOF on the BMMCs to study the immune landscape of these patient samples.

I have some comments about the study design, which I believe prevents definite conclusions given the heterogeneity of the induction regimens received by the patients and how this could affect the depth of response and MRD status.

- First, these samples are from PBMCs of the MM patients, which are mainly depictive of systemic immune repertoire, and we don't know how similar or different they are from the BMMCs of these patients. I understand that the authors performed CyTOF on samples from 13 patients and propose they are similar. However, it is a known issue for the bone marrow samples that they could be hemodiluted in such settings. Could the authors comment and address this, as I haven't seen a sufficient mention of this point?

We thank the reviewer for this comment. The objective of our study was to profile the immune system as comprehensively possible. Most investigations of the MM immune microenvironment have been limited to the bone marrow. Our study is unique in that we studied the immune cells within both bone marrow and peripheral blood. In doing so, we find both similarities as well as differences that would have been missed had we restricted our analysis to a single compartment. Furthermore, our observations comparing bone marrow and peripheral blood samples are consistent with a recent paper by Dr. Ghobrial's group (Zavidij et al., 2020)

We acknowledge that there is a possibility that there may be some circulating blood cells that contaminated the bone marrow aspirates making it challenging to discern a blood cell from a bone marrow cell. However, to our knowledge, there is no technique that can eliminate this effect completely and as a result this is a limitation of all studies of the bone marrow microenvironment. To minimize this effect, we used first-pull bone marrow aspirates for all MRD assessments which have been shown to have a lower degree of hemodilution (Rawstron et al., 2008). We have updated the results section to make note of this potential limitation (page 11).

- Second, I have a problem with the authors' definition of sustained vs. unstained MRD-ve. If patients with unsustained MRD-ve never been -ve from the beginning, then why were they included? They do not meet the criteria of MRD -ve state. This creates a misunderstanding of the results. I advise changing the term to something like successful vs. failed MRD state or any other accurate definition as sustained implies that a measure was already achieved and continues to occur, which is not the case here.

To avoid confusion when referring to study participants who did not achieve sustained MRD negativity, we have changed the term "unsustained MRD negative" to "non-

sustained MRD negative” through the manuscript and figures.

- In Table 1, the authors performed a chi-square test on categorical values, which is not the most appropriate test to be used but instead fisher exact given the small sample and frequency of events. The authors should use fisher's exact test to get an accurate p-value. The authors should also include an analysis of the difference between patients who received IMids and those who did not. Same for carfilzomib, which is known to be a more potent PI than bortezomib.

We thank the reviewer for this excellent suggestion and have updated Table 1 with P values from the Fisher exact test.

We also updated Table 1 to include induction therapy.

- In the unsustained MRD negative group, 10 patients had persistent MRD positivity from the start of maintenance and 2 patients who converted from MRD negative to positive within the first year of maintenance. However, in supplemental table 1. This data does not match with that in the main text.

Supplemental table 1 is correct while the text should read: “there were 9 patients who had persistent MRD positivity...”. The manuscript has been updated with this correction (page 5).

- In Figure 1 A, 2 patients became MRD +ve in the sustained MRD group; why were they not included in the unsustained group? How was the cutoff determined, and why? How did these patients do in terms of disease progression?

We believe the reviewer is referring to Figure 1B. This shows two patients (67 and 45) who lost MRD negativity after 2 years of maintenance therapy (notated by black triangles). They are correctly identified as achieving sustained MRD negativity because they had two consecutive MRD negative measurements one year apart. Both patients continue to remain on lenalidomide maintenance and have not progressed according to IMWG criteria.

Based on our observations from the previously published clinical trial for which this correlative study is based, we found that patients who achieve two consecutive MRD negative measurements at one year apart have the best overall outcome compared to those who achieve only one MRD negative measurement (Diamond et al., 2021). This finding has also been reported in other studies (San-Miguel et al., 2021).

- I see the sensitivity of the MRD testing was 10^{-5} , which is less sensitive than the current threshold. The authors need to acknowledge this limitation in the discussion and explain its potential relevance as it could have identified more MRD-positive patients.

We have revised the discussion to acknowledge that had we used a more sensitive MRD assay, some of the study participants may not have been classified as achieving sustained MRD negativity (page 15).

- On page 8, line 213: "Likewise, CD4 central memory and effector memory T cells were the most abundant T cell types on average in patients with unsustained MRD negativity

and no prior transplant (Fig. 4B)" While in page 6, the authors mention: "In contrast, a higher CD4:CD8 ratio was observed for all other comparisons, especially in patients with sustained MRD negativity without prior HDM-ASCT where CD4 central memory and naïve CD4 T cells were the most abundant lymphocyte." Could the authors comment on this part, as it looks contradictory, and discuss the significance and role of the central memory CD4 cells in the immune microenvironment?

The phrase on page 6 "where CD4 central memory and naïve CD4 T cells were the most abundant lymphocyte" is in relation to patients with sustained MRD negativity without prior HDM-ASCT. In contrast, the phrase on page 8 "CD4 central memory and effector memory T cells were the most abundant T cell types on average in patients with unsustained MRD negativity and no prior transplant" is in relation to all other patients. To avoid confusion, the phrase on page 6 has been revised to clarify this distinction.

We have added a statement in the results section to explain the differences in effector and central memory T cells (page 9) and highlighted a specific difference in clustering and gene expression unique to central memory CD4 T cells in patients with sustained MRD negativity (page 7).

- On page 9, line 232, the title is "T cell receptor β repertoire of the bone marrow and blood." I don't see anything on bone marrow in this paragraph. Why is that?

We apologize for this mistake. This was a typographical error which has been corrected in the revised manuscript (page 10).

- The authors discussed the correlation between their analysis of PBMCs and external cohorts of BMMCs in normal and MGUS patients. I see that the comparison is inherently limited given that these are two different tissues and the impact of a batch effect in such analysis. The latter could be accounted for, but the former is hard to account for in such a setting, and I don't know how reliable this comparison could be in deriving any conclusions. Moreover, the authors mentioned that normal and MGUS samples were more correlated to patients without a transplant in this study. What was the statistical test done, and was it significant to lead to this conclusion?

We agree with the reviewer that comparison of bone marrow and blood does have its limitations. For this reason, we directly compared the frequency of immune cells in bone marrow and blood collected on the same day from 4 patients. This analysis revealed no significant differences between the frequency of B cells, T cells, myeloid cells, NK cells, or dendritic cells in the blood compared to the marrow (Supplemental Figure 15). This suggests that the peripheral blood can approximate the frequency of immune cells in the bone marrow. To further validate this claim, we have performed a separate analysis comparing the BMMC using scRNAseq from the external cohort (Zavidij et al., 2020) to BMMC analyzed by CyTOF from patients in our study (Supplemental Figure 16). This analysis shows similar findings as we reported in Figure 6A-D.

In the revised manuscript, we have clarified that the statistical method we used to compare the frequency of immune cells in healthy individuals, precursor MM, and untreated MM to patients in our study was the Pearson correlation coefficient. To make more apparent which cell types contribute most to the similarity and differences with healthy bone marrow, we included a scatter plot (Figure 6E). Finally, we performed a linear regression analysis and reported the r^2 and p values, complementing the initial

Pearson results.

- The authors did not mention in their discussion the finding that in both the bone marrow and blood, there is an abundance of classical and non-classical monocytes, which are shown in multiple studies to be immunosuppressive, in patients with no prior transplant. The authors should also comment and discuss this and review the current literature on this topic to present the complete picture.

We have added an additional paragraph in the discussion addressing our findings regarding classical and non-classical monocytes as well as provided a brief literature review (page 14).

- I feel that the authors are leading their conclusions to the point that HSCT is not as effective and has a deleterious effect on the immune microenvironment. The authors need to mention the fact that randomized clinical trials like the IFM and Forte have shown that HSCT is still providing better outcomes in terms of PFS, the same surrogate the authors used here. It is essential to provide these results with what they also propose.

Immunosuppression is a well-known adverse effect of high-dose melphalan autologous stem cell transplant (HDM-ASCT). Our results indicate that the immune system continues to deviate from normal years after transplant. We hypothesize that this may translate into less effective response to immunotherapy administered at the time of relapse.

Despite these observations, we do not intend to make the claim that HDM-ASCT is ineffective or that its risks outweigh any benefits. We have updated the discussion and cited the findings from the IFM 2009, Forte, and DETERMINATION trials showing improved progression-free survival in patients who receive upfront transplant.

- Finally, the authors should have complemented this with genomic and transcriptomic data that they probably have on the CD138+ tumor cells or from FISH data. It will give a more comprehensive picture rather than this divided approach on the predictors of MRD status. I believe this is an essential part to be added to the data.

We agree with this reviewer that genomic analysis of the tumor cells would improve our understanding of the mechanisms of MM disease progression during maintenance therapy. However, since all patients enrolled in our clinical study had completed induction therapy and because many of them had achieved MRD negativity, there was insufficient tumor cells to perform this analysis.

Reviewer #2 (Remarks to the Author):

In this manuscript Coffey et al. profiled the immune system of patients with newly diagnosed myeloma receiving lenalidomide maintenance with the main objective to correlate long term treatment response with specific immune changes. By performing CyTOF, single cell RNA sequencing and TCR analysis in longitudinal collected samples before and 1 year after the initial treatment the authors observed that the composition of the T cell population in the bone marrow of these patients somehow varies in patients achieving sustained minimal residual disease (MRD) negativity versus patients who never achieved or were unable to maintain MRD negativity.

The authors found a higher increase in terminal effector CD8+ T cells resulting in a reduced CD4+:CD8+ ratio in the marrow of patients maintaining long term MRD negativity. While the study is somehow interesting, I believe that the scientific conclusions cannot be fully supported from the presented results. It is extremely well known that lenalidomide mechanisms of action is to increase cytotoxic T cell expansion and that once MM cells are eradicated the BM immune-compositions better resembles the one found in cancer free donors. This is also somehow represented from the fact that the authors reported that CD27- non-memory B cells was found the dominant cell type at baseline in patients achieving sustained MRD negativity who had a prior transplant, while this finding was not seen in other subgroups further. From the presented data it seems like that a healthier BM composition maybe essential to maintain MRD negativity upon Lenalidomide treatment. In my point of view the authors should further focus on understanding how the initial response to lenalidomide (early cycles) of the immune environment can correlate with longer MRD negativity. This approach could be extremely informative to understand the ability of patients to deeply respond to the treatment based on the immune changes more than on the plasma cell clone diversity. Specifically:

1. Did the authors collect peripheral blood or bone marrow samples at earlier time points to assess whether initial response to lenalidomide treatment associated with terminal effector CD8+ T cell expansion, NKT can correlate with prolonged MRD negativity? In my point of view the analysis of early time points could be more informative about lenalidomide mechanisms of action and the importance of the microenvironment response to keep the tumoral cell under control, independently of their initial genetic profile. Unfortunately, after 1 year of treatment it is not clear if the presence/absence of CD8+T cells and changes in the other immune subsets is the cause or only the consequence of long MRD negativity. It is also not clear of how this information can help understanding the role of the environment to keep disease under control.

Bone marrow biopsies were obtained annually from all participants enrolled in on our clinical trial. We did not collect bone marrow samples less than one year after initiation of lenalidomide maintenance therapy. We believe that if we profiled samples within the first few months of starting lenalidomide, we would have observed a more substantial effect of the preceding induction therapy, especially autologous stem cell transplant. To minimize the influence of induction and allow sufficient time for immune reconstitution to occur post-transplant, we chose to study samples one year after maintenance.

We agree that it is difficult to discern whether the differences in immune cell subsets we detected are caused by or a consequence of sustained MRD negativity. Nevertheless, our results do inform us of the state of the immune system during the first remission which have important implications for the use of immunotherapy at first relapse. Importantly, we find that MRD positivity continues to influence the frequency and

phenotype of immune cells that may potentially undermine the effectiveness of immune therapy. This observation is important because it may theoretically support using lymphodepletion reduce immunosuppressive cells prior administration of immune targeted therapy in patients with relapsed disease.

2. Is the composition of the immune environment changes once patients actively progress?

Only patients with non-sustained MRD negativity experienced disease progression (Figure 1D) enabling us to observe changes in the immune system just before progression occurred. These are the findings reported in the manuscript. In contrast, we could not assess changes immune microenvironment among patients with sustained MRD negativity since none of the patients progressed.

3. The legends of all the supplementary figures are not clear and it is extremely hard to follow what the authors want to show in the associated Figures.

We thank the reviewer for this comment. To ensure clarity of the figures, we have revised the supplemental figure legends to state the purpose of the figures more clearly.

Overall, this is an interesting manuscript exploring the importance of the immune environment in response to Lenalidomide maintenance therapy, but the novelty is somehow compromised from the limited time points analyzed.

We agree with the reviewer that our sample size limits the statistical power for these high-dimensional analyses. Indeed, this is a dilemma for the field at large. The current study is comparable in scale with prior single-cell studies which reflects the limited number of available samples from clinical trials partnered with cost-prohibitive assays. We have discussed the impact and power of this study with experts on the field and although we are not powered to make populational claims regarding disease, we carefully report significant observations of the immune composition and molecular signatures in MRD negative patients under a specific treatment. The current study was supported by a large clinical trial budget partnered with a large multi-center immunotherapy grant from the Multiple Myeloma Research Foundation. More than \$1 million was used to support this effort and yet the funding did not allow a larger number of samples to be evaluated. Going forward, more affordable technologies will facilitate larger patient cohorts and in turn improve statistical power.

Reviewer #3 (Remarks to the Author):

In this manuscript Coffey et al present data about the impact of immune microenvironment on MRD negativity achievement. The project is novel, but I have some major concerns:

Novelty is based on the methodology employed, but there is some defects in the design to get their objective. The results do not support completely the conclusions.

1. The MM population studied is very heterogenous and the treatments employed are assorted. This could make the conclusion weaker.

Our findings represent “real world” patients in whom induction therapy consisted of combination of proteasome inhibitors, immunomodulatory drugs, and low dose steroids, with or without subsequent autologous stem cell transplant. All patients at the time of sample collection were receiving the same maintenance therapy. Furthermore, we chose to study patients one year after starting maintenance therapy to minimize the impact of the preceding induction therapy. To account for the profound affect of autologous stem cell transplant on immune microenvironment, we analyzed these patients separately. Also, intensive quality control analysis was performed on the samples to ensure high quality data.

2. The number of patients analyzed is small to get strong conclusions 12 and 11

To our knowledge, this is the largest single-cell analysis to date of the immune microenvironment in patients with MM undergoing maintenance therapy. We acknowledge that we lacked sufficient statistical power to uncover differences between our subgroups and have stated this limitation in the discussion (page 15). Nevertheless, our findings generate novel hypotheses that can be validated in larger studies. To ensure accurate conclusions, the results were reviewed multiple times by independent analysts and each sample was treated as independent observations to ensure the highest quality of interpretation of the data as possible.

3. Probably the hypothesis should be reformulated to compare patients that relapse early and those that relapsed late. Because some patients achieving MRD negativity relapse

This is indeed how our study was designed. We compared patients with sustained MRD negativity (median progression-free survival > 4 years, Figure 1D) to patients with non-sustained MRD negativity (median progression-free survival 1 year, Figure 1D). We chose not to group patients based on their initial MRD status since, as the reviewer points out, achieving MRD negativity at a single time point is less prognostic than sustaining MRD negativity over multiple time points.

4. Should be of interests to study a group of patients with MM in CR but without treatment with lenalidomide to define the effect of lenalidomide in the microenvironment

Although we compared our findings with 9 healthy donors, 5 patients with MGUS, 11 patients with SMM, and 7 patients with untreated MM (Zavidij et al., 2020), we did not compare to patients with MM in complete remission who are not receiving lenalidomide. While we agree that would be a good control population to identify the effects of lenalidomide, that is outside the scope of our original research plan and is not currently feasible since it would require substantial additional resources.

Reviewer #4 (Remarks to the Author):

Summary

This is a well written paper addressing an interesting question around the contribution of immune function to quality of disease response in treated myeloma patients, and is based on the analysis of a nice and novel dataset. The authors have undertaken single cell analysis of bone marrow and blood cells from patients with MM undergoing lenalidomide maintenance following initial therapy, employing a variety of tools: single cell RNA sequencing, V(D)J sequencing and TCRbeta sequencing of peripheral blood mononuclear cells and CyTOF analysis of bone marrow mononuclear cells. Each patient was sampled at start of maintenance, and then subsequently after a year of maintenance. The aim of the study was to uncover immunological features associated with sustained MRD negative response in patients receiving Lenalidomide maintenance. 12 patients had sustained MRD negativity, while 11 were either MRD positive throughout, or converted from MRD negative to positive within the year. Patients had received a variety of previous therapies but the strongest influence on their immune profiles was the receipt of high dose therapy and stem cell transplantation (SCT), that resulted in distinctive T cell features which persisted a year after maintenance. The authors report higher frequencies of naïve and central memory CD4 cells in patients without a prior history of transplant who had sustained MRD negativity, however no statistical information is given.

Major issues

Patient groups.

As acknowledged by the authors, the 2 groups of patients were unbalanced in several aspects: those with sustained MRD negativity were slightly younger, were less likely to have high risk features such as high risk genetics and ISS3, and less likely to have received SCT. They were also more likely to have received carfilzomib as part of their induction regimen (8/12) compared with the MRD positive group, of whom only 1 received a carfilzomib regimen as first line, while another received it at 4th line before proceeding to SCT. Thus the achievement of MRD negative response in the first place was influenced by disease features and treatment received, and the ability to sustain this quality of response with lenalidomide maintenance may have less to do with immune 'fitness' than with prior treatment history. In order to answer the question that the authors posed, ie. the immunological determinants of sustained MRD negative response, they would have to start with similar patient populations at a similar level of disease response, then compare those that sustained MRD negativity with those that didn't. Given the patient cohort that they have, however, the authors should at least acknowledge the possible confounding effect of the treatment imbalance on their results.

We have updated Table 1 with statistical testing to clearly reveal the identified differences in our subgroups. We have also added a statement to the discussion acknowledging that differences in disease biology and treatment history may have confounded our ability to reveal differences in the immune microenvironment (page 15).

Results

Figure 1 is This is an excellent schematic outlining the cohort and patient datasets.

Single cell RNA sequencing data and Figure 2.

The authors have used a reference based mapping tool provided by Azimuth, and have omitted a gene expression clustering step in their scRNAseq analysis. By relying on the cell identities predicted by Azimuth the authors may miss groups of cells which may be enriched in specific conditions. For example, if a group of Treg cells are enriched in MRD+ versus MRD- patients, but all are uniformly given the label "Treg" by Azimuth, this observation would be lost. If

however, Tregs were subjected to clustering analysis and these two groups of cells formed distinct clusters, this observation could be noted.

This is an excellent recommendation, and we thank the reviewer for this helpful comment. We have performed an additional analysis whereby sustained and non-sustained MRD negative samples are compared within each cell type (Supplemental Figures 6-11). This revealed that for some cell types, there are subpopulation of cells that cluster separately according to the MRD negative status. For example, there is a group of central memory CD4 T cell that are specific to patients with sustained MRD negativity and no prior transplant (Supplemental Figure 7). We subsequently performed differential gene expression analysis to identify that *EGR1* expression is significantly reduced in central memory CD4 T cell in patients with sustained MRD negativity and no prior transplant. *ERG1* has been shown to induce T-bet transcription which is a Th1-specific transcription factor that is directly involved in TCR signaling and the IFN-gamma-STAT1 and IL-12-STAT4 pathways (Shin et al., 2009). We have updated the results section to reflect these changes.

Furthermore, the authors assume the phenotypic spectrum occupied by PBMC in their data is encompassed by Azimuth (which will only predict labels present in its reference), which may not be most appropriate.

The advantage of referenced based mapping for the purposes of cell phenotyping is that it removes the subjectivity from manually assignment based on only a handful of markers. Since there are certain cell subsets (e.g. exhausted T cells) that are not labeled in our reference, we derived a separate technique for identifying them (see description below).

The author's decision to remove cell types present <200 times in the dataset (line 399) also seems needlessly restrictive.

Below is a table cell types removed and counts per sample. The rationale for removing the cells is that there were too few to be able to perform statistical comparisons with meaningful results due to insufficient power.

Cell type	Sustained MRD-		Non-sustained MRD-	
	No prior ASCT-HDM	Prior ASCT-HDM	No prior ASCT	Prior ASCT-HDM
AXL+ dendritic cells	0.021%	0.056%	0.027%	0.038%
Conventional Type 1 Dendritic Cells	0.167%	0.019%	0.055%	0.069%
Double negative T cell	0.135%	0.113%	0.197%	0.205%
Erythrocyte	0.079%	0.038%	0.016%	0.081%
Innate lymphoid cells	0.132%	0.056%	0.071%	0.055%
Progenitor cells	0.177%	0.019%	0.109%	0.052%
Proliferating CD4 T cell	0.148%	0.075%	0.098%	0.133%
Proliferating CD8 T cell	0.098%	0.169%	0.136%	0.148%
Total cell count	37,766	5,320	18,318	34,557

There was also no differential expression analysis presented, which could easily occupy an entire figure by itself given the number of samples available for each group of patients. We

would strongly recommend the Pseudobulk approaches outlined in Squair et al. (<https://doi.org/10.1038/s41467-021-25960-2>) for such analysis,

This is another excellent recommendation by the reviewer and we have carried out this analysis and reported our findings alongside the recommended subcluster analysis (Supplemental Figures 6-11). For this analysis we used the EdgeR-LRT pseudobulk method from the Libra R package (Squair et al., 2021).

Figure 2A - B. It is unclear why trajectory analysis was performed, and it adds little to the manuscript.

Single-cell trajectories were depicted to show the cell lineage relationship in addition to their gene expression clusters and make it easier to compare clusters across different UMAP projections.

Figures 2C - E and Fig S1 do not effectively show quantitative statistical differences between the patient groups at a per-patient level. Boxplots of proportions, with each point being a donor, would be more suitable.

While we chose to use area plots to depict change over time, we also agree that is useful to also show the distribution of cell frequencies within subgroups. In the revised manuscript we have included box plots comparing the cell frequencies across all cell types for baseline and follow-up samples aggregated by prior history of transplant (Supplemental Figures 4 and 14).

It is unclear how genes were selected in Fig S2. Suggest display marker genes derived by differential expression.

The plot showing the top genes most specifically expressed by each cell type was generated using the `top_markers` function from the `monocle3` R package (Cao et al., 2019). Further filtering of genes was performed to include those expressed by 10% or more of cells. The gene with the highest pseudo R-squared value, a measure of how well the gene expression model fits the categorical data relative to the null model, are shown in the figure. The figure caption has been updated to include this additional information.

Is the T cell clonality presented in Fig S3 correct? Why are naïve CD4 and CD8 T cells clonotypes, biologically non-clonal, shown as expanded? More information could be acquired by examining clonality beyond a binary >1 threshold. Defining deciles or larger expansion, for example as >10 cells, could help elucidate if any T cell subsets are particularly expanded.

We have updated Supplemental Figure 3 using a gradient to depict VDJ clonality rather using “expanded” and “non-expanded” classes. This more accurately shows that naïve CD4 and CD8 T cells have relatively few clonotypes compared to terminal effector CD8 T cells.

Figure 3 and T cell exhaustion

There is insufficient information to substantiate the authors' claims to have identified exhausted T cells as outlined in Figure 2. The identification of exhaustion based on (co-)expression of single markers, each assessed in a binary fashion, can be misleading; the hallmark of an exhausted T cell is the prolonged co-expression of several co-inhibitory receptors, each of which may not be significantly expressed. The authors have identified 3 markers, but ICOS, LAG3 at least are also activation markers. This is in keeping with the higher expression on

proliferating T cells, again this is also seen in exhausted subsets and is simply an indication that the cells are responding to a stimulus.

Several key transcriptional markers of these cells, including PD1(PDCD1) or transcription factors associated with exhaustion like LAYN or TOX/TOX2 seem uninvolved or were not investigated. An exhaustion program is most likely to be detected in cell types with a phenotype resembling effector T cells, hence the absence of exhaustion markers in the Azimuth-predicted effector populations is noteworthy. Of those exhaustion-associated markers shown to be expressed, some of the highest expression being in MAIT and DN T cells (which are not canonically associated with the exhaustion pathway) might be a warning sign that these genes are being expressed absolutely at very low levels. The expression of a subset of exhaustion markers in proliferating T cells is promising, but the authors should note the higher sequencing depth achieved in proliferating cells will boost the number of detected genes and give an impression of higher per-gene expression (making the absence of PDCD1 in these cells even more noteworthy). Further, these markers may be expressed in proliferating CD4 because this includes proliferating Tregs.

Fig 3B should be re-done as a heatmap including classical transcription factors associated with exhaustion such as Tbet, TOX and LAYN.

Fig 3C: there is no definition as to what discriminates cells which are or are not exhausted (ie. a threshold or score)

These are excellent points by the reviewer and we have revised our definition of exhausted T cells. Previously, we considered only cells expressing one or more inhibitory receptors. In the revised definition, we also included the cytotoxic molecule Granzyme B and transcription factors ID2 and BLIMP1 (TOX was not significantly expressed in our dataset). Our current definition now states that T cells were subclassified as exhausted if they express all the following genes: *GZMB*, *ID2*, *BLIMP1*, and one or more inhibitory receptors (*TIM3*, *CD244*, *CD39*, *CD160*, *CD200*, *LAG3*, *TIGIT*, or *CTLA4*). This definition is based on the key molecules that have been reported to be consistently expressed within terminally exhausted T cells (Kallies et al., 2020). To further evaluate this definition, we performed a diffusion map analysis as shown in revised Figures 3A and 3B. This reveals that T cells labeled as exhausted exhibited a gene expression program most closely related to terminal effector CD8 T cells. With this new definition, we have revised Figure 3C showing an increase in terminally exhausted CD8 T cells from baseline to follow-up in patients with non-sustained MRD negativity with or without transplant. In contrast, terminally exhausted CD8 T cells appeared to decrease in patients with sustained MRD negativity and prior transplant or no change in patients without prior transplant.

TCR sequencing analysis

The analysis of TCR diversity is problematic unless the authors undertook subsampling to a fixed number of UMI defined TCR mRNA sequences? This is required in order to account for the wide variation in the actual number of TCR sequences obtained from each sample.

Unlike complementary DNA or mRNA measurements that are obscured by cell expression, we performed our TCR analysis on genomic DNA extracted from the blood using the ImmunoSeq assay (Adaptive Biotechnologies). As a result, reads can be quantified providing an absolute cell count that does not require unique molecular identifiers. This makes it possible to accurately assess clonal expansion and tissue density of T cells (Robins et al., 2009).

The packing diagrams (Fig 3E), which clearly show the phenotype of the same clonotype are nice, however, the continuous “heat” colour map is harder to read than a selection of diverse colors.

Figure 3E has been updated using a discrete color palette.

Publicity does not mean shared reactivity. Defining a shared TCRB as “shared TCR specific group” is not strictly accurate and the specificity of these clonotypes can only be speculated about at this point. If the authors wanted to pursue this route, could it be shown that such publicity in PB TCRB chain does not exist in age-matched healthy donors? If so, was it associated with a particular T cell phenotype in this dataset? Furthermore, it should be asked if any TCRs, particularly the TCRB chains shared between donors, were also found annotated in public TCR databases such as the VDJdb (<https://vdjdb.cdr3.net/>). Beyond a shared reactivity, TCR chains may be frequently identified because they possess a higher probability of generation (not every TCR chain has a similar likelihood of being generated during T cell development, with some being much more frequent than others). Perhaps the tool OLGA (<https://github.com/statbiophys/OLGA>) and associated manuscript could be employed here.

These are excellent points made by the reviewer and we have incorporated the recommended analyses into the revised manuscript (page 11) and report the data in Supplemental table 3. The underlying goal of the T cell receptor β analysis was to identify differentially abundant CDR3 β amino acid sequences in patients with either sustained or non-sustained MRD negativity. To further investigate these differentially detected sequences we queried them in the VDJdb, McPAS-TCR, PIRD TBAdb, and LymphoSeqDB databases to reveal known antigen specificities. This uncovered 14 CDR3 β amino acid sequences with known antigen specificity to a variety of microbial antigens and one human antigen. We also searched for the CDR3 β amino acid sequences within our scRNAseq dataset to determine if the receptor was associated with a particular T cell phenotype. Due to the shallow depth of single-cell sequencing, this revealed only 4 CDR3 β amino acid, limiting our ability to draw further conclusions. To determine the prevalence of the CDR3 β amino acid sequencing among 55 healthy individuals ages 0-90 years, we searched for the sequences in LymphoSeqDB database and reported the frequency of healthy individuals who share this same sequence. Finally, we used OLGA to compute the generation probabilities. Together, these analyses revealed that T cell receptor β sequences that are unique to patients with sustained MRD negativity are infrequently found in healthy individuals and have a relatively low CDR3 sequence generation probability. This suggests that these sequences are unlikely to be shared because they are prevalent in the human population or have a high probability of sequence generation.

CyTOF data, Figure 4

This is the only data to be obtained from BM samples, and the authors acknowledge that there are differences in subsets, but fail to match the cell populations in CyTOF those in the scRNA seq dataset. The CyTOF analysis would benefit from more exploratory clustering of the scRNAseq data. The interesting compositional findings from the CyTOF regarding NKT and CDC cells cannot be supported by the scRNAseq as these cell types were absent.

We performed hierarchical clustering and tSNE analysis of CyTOF data and reported our findings in Figure 4A and Supplemental Figure 13 respectively. The heatmap in Figure 4A shows there are 3 major clusters of patients enriched in either classical monocytes, conventional dendritic cells, or CD16⁻ natural killer cells. However, these clusters do not clearly

associate with MRD- status, time points, or history of transplant.

Fig 4B, C, D. Same comment as for Fig 2C, D, E. Can the authors show stats for the CD8 effector subset frequency in patients with prior SCT?

Please see response to Figure 2C-E above.

Fig 4B. The authors state that the overall proportion of T cells is higher in BM of patients with sustained MRD negativity and no SCT and in those with unsustained MRD negativity and prior SCT but surely it is the other way around?

That is correct, this is a typographical error and the manuscript has been updated.

In Fig 4E it looks as if Treg cells are higher in patients with sustained MRD negativity at follow up, whilst the single cell sequencing data in Fig 2H suggests that Treg cells actually reduce in these patients. Can the authors explain this apparent discrepancy in their findings?

These are two separate analyses on separate tissues. The analysis in 2H is evaluating the percent change in circulating regulatory T cells whereas the analysis in figure 4E is the differential abundance of bone marrow cells at baseline (right panel) and follow-up (left panel) time points.

Fig 5. The same caveats regarding public TCR beta chains described in Figure 3 apply to Fig 5C.

Please see response to Figure 3 above.

Figure 6. Comparison with healthy donors and precursor conditions.

The correlation between patients achieving MRD status and healthy donor marrows is interesting (although Fig 6B could require some per-donor statistics). As this is presented as one of the major findings of the manuscript, further analysis is required.

To make more apparent which cell types contribute most to the similarity and differences with healthy bone marrow, we performed a linear regression analysis and included a scatter plot (Figure 6D). From this visualization it can be more clearly seen which cell types share similar frequency to healthy individuals compared to those cell types that do not.

Fig 6C: Presume this plots median Pearson's correlation comparing disease group with healthy bone marrow? Given the wealth of sequencing data available, surely a more indepth analysis could yield information on whether if this is dominated by a subset of cell types which could serve as individual biomarkers vs whole bone marrow scRNAseq?

Machine learning approaches (feature selection) can determine which variables (here, cell types) contribute most to an outcome (here, correlation). Otherwise the value and relevance of this comparison is questionable.

Please see response to figure 6 above.

Discussion

This largely re-iterates the descriptive narrative of the results section and provides no biological interpretation or insight into the clinical or otherwise significance of the findings.

The authors attempt a summary of the immune correlates of sustained MRD negativity in non-

SCT and SCT patients, observing an increase in circulating CD4 cells, less exhausted T cells in non-SCT patients while in SCT patients had more circulating CD8 effectors and T cells with exhaustion phenotype. Aside from the lack of statistical data to back them up, these findings do not inform on the possible function of lenalidomide, or whether these immune correlates are as a result of the low disease burden, allowing a return to a more healthy immune system, or whether they indicate some sort of immune contribution to disease control.

This is an important point and we have added language to the discussion in the revised manuscript to address this (page 15).

The authors state there that “..history of HDT-ASCT was associated with a increased diversity before the onset of maintenance...”, this is not what they report?

This is a typographical error and we have updated the manuscript to read: “no history of HDT-ASCT was associated with an increased diversity before the onset of maintenance” (page 13).

The authors need to differentiate, if they can, which immune features are indicative of sustaining an MRD negative response, from those that indicate an effect of lenalidomide therapy. The heterogenous nature of their patient cohort makes this problematic. Suggest that they focus on one cohort, preferably the non-ASCT cohort, where the immune landscape is not so skewed by the history of the ASCT.

Through comparison of baseline samples (pre-lenalidomide), we compared immune features of sustained versus non-sustained MRD negativity. The findings in the bone marrow are shown in Figures 2F (left panel) for blood and 4E (left panel) for bone marrow.

REFERENCES

- Cao, J., Spielmann, M., Qiu, X., Huang, X., Ibrahim, D. M., Hill, A. J., Zhang, F., Mundlos, S., Christiansen, L., Steemers, F. J., Trapnell, C., & Shendure, J. (2019). The single-cell transcriptional landscape of mammalian organogenesis. *Nature*, *566*(7745), 496–502. <https://doi.org/10.1038/s41586-019-0969-x>
- Diamond, B., Korde, N., Lesokhin, A. M., Smith, E. L., Shah, U., Mailankody, S., Hultcrantz, M., Hassoun, H., Lu, S. X., Tan, C., Rustad, E. H., Maura, F., Maclachlan, K., Peterson, T., Derkach, A., Devlin, S., Landau, H. J., Scordo, M., Chung, D. J., ... Landgren, O. (2021). Dynamics of minimal residual disease in patients with multiple myeloma on continuous lenalidomide maintenance: a single-arm, single-centre, phase 2 trial. *The Lancet Haematology*, *8*(6), e422–e432. [https://doi.org/10.1016/s2352-3026\(21\)00130-7](https://doi.org/10.1016/s2352-3026(21)00130-7)
- Kallies, A., Zehn, D., & Utschneider, D. T. (2020). Precursor exhausted T cells: key to successful immunotherapy? *Nature Reviews Immunology*, *20*(2), 128–136. <https://doi.org/10.1038/s41577-019-0223-7>
- Rawstron, A. C., Orfao, A., Beksac, M., Bezdicikova, L., Brooimans, R. A., Bumbea, H., Dalva, K., Fuhler, G., Gratama, J., Hose, D., Kovarova, L., Lioznov, M., Mateo, G., Morilla, R., Mylin, A. K., Omedé, P., Pellat-Deceunynck, C., Andres, M. P., Petrucci, M., ... Network, E. M.

(2008). Report of the European Myeloma Network on multiparametric flow cytometry in multiple myeloma and related disorders. *Haematologica*, 93(3), 431–438. <https://doi.org/10.3324/haematol.11080>

Robins, H. S., Campregher, P. V., Srivastava, S. K., Wachter, A., Turtle, C. J., Kahsai, O., Riddell, S. R., Warren, E. H., & Carlson, C. S. (2009). Comprehensive assessment of T-cell receptor β -chain diversity in $\alpha\beta$ T cells. *Blood*, 114(19), 4099–4107. <https://doi.org/10.1182/blood-2009-04-217604>

San-Miguel, J., Avet-Loiseau, H., Paiva, B., Kumar, S., Dimopoulos, M. A., Facon, T., Mateos, M.-V., Touzeau, C., Jakubowiak, A., Usmani, S. Z., Cook, G., Cavo, M., Quach, H., Ukrepec, J., Ramaswami, P., Pei, H., Qi, M., Sun, S., Wang, J., ... Bahlis, N. J. (2021). Sustained minimal residual disease negativity in newly diagnosed multiple myeloma and the impact of daratumumab in MAIA and ALCYONE. *Blood*, 139(4), 492–501. <https://doi.org/10.1182/blood.2020010439>

Shin, H.-J., Lee, J.-B., Park, S.-H., Chang, J., & Lee, C.-W. (2009). T-bet expression is regulated by EGR1-mediated signaling in activated T cells. *Clinical Immunology*, 131(3), 385–394. <https://doi.org/10.1016/j.clim.2009.02.009>

Squair, J. W., Gautier, M., Kathe, C., Anderson, M. A., James, N. D., Hutson, T. H., Hudelle, R., Qaiser, T., Matson, K. J. E., Barraud, Q., Levine, A. J., Manno, G. L., Skinnider, M. A., & Courtine, G. (2021). Confronting false discoveries in single-cell differential expression. *Nature Communications*, 12(1), 5692. <https://doi.org/10.1038/s41467-021-25960-2>

Zavidij, O., Haradhvala, N. J., Mouhieddine, T. H., Sklavenitis-Pistofidis, R., Cai, S., Reidy, M., Rahmat, M., Flaifel, A., Ferland, B., Su, N. K., Agius, M. P., Park, J., Manier, S., Bustoros, M., Huynh, D., Capelletti, M., Berrios, B., Liu, C.-J., He, M. X., ... Ghobrial, I. M. (2020). Single-cell RNA sequencing reveals compromised immune microenvironment in precursor stages of multiple myeloma. *Nature Cancer*, 1–14. <https://doi.org/10.1038/s43018-020-0053-3>

REVIEWER COMMENTS

Reviewer #1 (Remarks to the Author):

I would like to thank the authors for revising their manuscript and addressing the comments I raised. One last comment is that in the results section in the updated manuscript, the authors mentioned that "In general, patients with sustained MRD negativity were more likely to be younger, lack high-risk features such as ISS III disease or high-risk cytogenetics, and not have received high- dose melphalan with autologous stem cell transplant (HDM-ASCT)".

I believe this sentence need to be revised or removed as per Table 1 results, the only statistically significant difference is the High-risk cytogenetics. So this needs to be fixed to reflect the results as it gives the idea that these were significant differences.

Otherwise, I believe the manuscript now is more clear and in better shape after acknowledging some of the limitations.

Reviewer #2 (Remarks to the Author):

In this revised manuscript Coffey et al. profiled the immune system in patients with newly diagnosed MM receiving continuous lenalidomide maintenance therapy with the aim of uncovering correlates of long-term treatment response. The authors used single-cell RNA sequencing and T cell receptor β sequencing of the peripheral blood and CyTOF mass cytometry of the bone marrow to characterize the immune landscape in 23 patients before and one year after lenalidomide exposure. The authors compared patients achieving sustained minimal residual disease (MRD) negativity to patients who never achieved or were unable to maintain MRD negativity. While I believe that the analyses that the authors are reporting can support the future understanding of the importance of the immune environment in keeping the myeloma cancer cells under control. I still don't believe that the information contained in this manuscript can clearly answer specific questions, beside the collection of many cutting age analysis in primary patients. If the authors could at least identify possible differences at baseline that can prospectively inform which group of patients can achieve long term MRD negativity versus others that would be important and increase the significance of their results. I still believe that the analysis at 1 year are not informative about the role of the immune system to maintain sustained minimal residual disease (MRD) negativity or to inform about the possible effect of lenalidomide. In my previous comments I asked if early response to lenalidomide including expansion of peripheral blood cytotoxic T cells could correlate with MRD negativity but this information have not been included in the revised form of the manuscript. This analysis could be also performed in a different set of patients. Overall this is a well written manuscript reporting very comprehensive correlative studies but the results are not particularly informative in the way they are reported.

Reviewer #3 (Remarks to the Author):

In this report Coffey analyze with innovative techniques de immune systems and impact on MRD and its sustainability. The manuscript is of merit however the main limitation is the small number of cases studies, this could compromise some results

Reviewer #4 (Remarks to the Author):

To the authors

Thank you for the detailed revision of the manuscript and for taken up several of my suggestions and performing additional analyses. While some new analyses are performed admirably and the resulting data contribute meaningfully to the manuscript (namely the TCR analysis), others have not done so. Several original comments have not been addressed convincingly.

Imbalance of patient groups

Thank you for including statistical information regarding the differences between the ASCT and no-ASCT group in Table 1. I could not find the “statement .. the discussion acknowledging that differences in disease biology and treatment history may have confounded our ability to reveal differences in the immune microenvironment”.

Differential gene expression analysis of single cell RNA sequencing data

Page 7, the speculation about relevance of ERG1 should be relocated to the discussion. I would caution against such extensive speculation for the differential expression of a single gene, seemingly chosen at random. Do the authors genuinely believe the activation of central memory T cells (defined by a single gene whose activity, namely induction of Tbet and STAT1, is not even shown) is significantly involved in MRD negativity? If so, how would the data support this?

Page 9, the sentence beginning “central memory T cells proliferate extensively and are...” is too interpretative for the results and should be moved to the discussion.

Definition of exhausted cells

Thank you for the additional analysis and the attempt to resolve exhausted cells. Unfortunately although the text describes exhaustion-associated genes, the actual expression of these genes is not shown. Furthermore, this classification of exhaustion is irregular. While the co-expression of GZMB, ID2 and BLIMP1 could describe exhausted cells, the selective expression of at least one of TIM3, CD244, CD39, CD160, CD200, LAG3, TIGIT, or CTLA4 casts a wide net regarding T cell phenotype. For instance, CD160 is expressed by NKT cells. Analysis of terminally exhausted cells by 10X scRNAseq will readily reveal cells with high-level co-expression of these checkpoint molecules, not only one. I note that TOX, a widely-used transcriptional marker of exhausted T cells (van der Leun, Thommen and Schumacher, 2020; Ren et al., 2021), was not significantly expressed in this dataset. Does this not suggest exhausted cells were not detected? And what of other single-cell RNAseq markers described in these reviews such as LAYN or CXCL13? Was the dataset at any point filtered to remove such genes? To be convinced that these genuinely resemble exhausted T cells, could the authors produce a per-cell heatmap (not a cluster-aggregated dotplot) and per-gene UMAPs showing the expression of the genes the authors used to classify exhausted cells (as well as LAYN and CXCL13), and show these cells regularly co-express these genes.

To re-iterate, some checkpoint molecules are also activation markers. Perhaps best to use a description of terminal effectors simply expressing markers of activation.

However, ignoring the question of identity, the differential abundance or expression analysis does not seem to support the relevance of these cells in this cohort; the differences in Fig.3C lack significance and the differential expression within CD8+ Terminal Effectors (within which terminally exhausted CD8 cells sit) in Fig.S7 do not reveal exhaustion markers. Given these problems with definition I am uncomfortable with inclusion of this subset in Fig.6E.

Figure 6: Comparison of immune cell frequencies in healthy donor and patient groups

The authors have not sufficiently taken into account the compositional heterogeneity amongst individual donors. Fig.6B-D appear to show correlations between donor-aggregated proportion for the different patient groups but this approach under-estimates the immune composition heterogeneity

within each group. Could a compositional clustering approach similar to Combes et al., (2022) be undertaken instead? Would such unsupervised compositional analysis reveal the same similarities between healthy and non-ASCT:Sus groups?

TCR sequence analysis

Thank you for undertaking additional analysis and the interesting finding that several CDR3 beta are unique to patients with sustained MRD negativity is convincing and represents a promising avenue of investigation in the manuscript. Could the authors follow up this finding in more detail? For instance, what is the phenotype of these clones in the scRNAseq dataset? Do they share structural features in the CDR3 region? Could these clones be used to derive a gene signature of prognostic relevance? I appreciate this may not be possible but could these sequences be generated in vitro and their reactivity assessed against autologous tumour material, similar to Caushi et al., (2021).

Discussion

This is improved however suggest to focus on the no-ASCT cohort as here the correlations of immune cell frequencies with sustained MRD appear more consistent, eg. naïve CD4 and CD4CM cells, while the effect of ASCT is so dominant that it is more difficult, at least within the analyses conducted so far, to discern a consistent narrative around sustained MRD response. Suggest also to discuss the clinical implications of the unique shared CDR3 clones amongst patients with sustained MRD negative response. Suggest also that unless able to more convincingly resolve exhausted cells, to tone down the reference to this subset, and alter the narrative in Fig 6E.

Suggest remove the sentence “HDM-ASCT exerts a significant, long-term effect on T cell phenotype and does not appear to alter the ability of patients to achieve a durable remission.” as our extensive clinical experience would suggest otherwise. Instead re-phrase to say something like “..the changes due to ASCT preclude any consistent finding regarding correlation of T cell phenotype with sustained MRD response within the current analysis”.

References:

- Caushi, J. X. et al. (2021) ‘Transcriptional programs of neoantigen-specific TIL in anti-PD-1-treated lung cancers’, *Nature* 2021 596:7870, 596(7870), pp. 126–132. doi: 10.1038/s41586-021-03752-4.
- Combes, A. J. et al. (2022) ‘Discovering dominant tumor immune archetypes in a pan-cancer census’, *Cell*, 185(1), pp. 184-203.e19. doi: 10.1016/J.CELL.2021.12.004.
- van der Leun, A. M., Thommen, D. S. and Schumacher, T. N. (2020) ‘CD8+ T cell states in human cancer: insights from single-cell analysis’, *Nature Reviews Cancer. Nature Research*, pp. 218–232. doi: 10.1038/s41568-019-0235-4.
- Ren, X. et al. (2021) ‘Insights Gained from Single-Cell Analysis of Immune Cells in the Tumor Microenvironment’, <https://doi.org/10.1146/annurev-immunol-110519-071134>, 39, pp. 583–609. doi: 10.1146/ANNUREV-IMMUNOL-110519-071134.

Reviewer #5 (Remarks to the Author):

This work studied sustained MRD negativity in patients with multiple myeloma using 3 single-cell techniques to longitudinally characterized the immune landscape in 23 patients before and one year after lenalidomide exposure. Study design, hypothesis generation, and clinical samples are great; however, data analysis and conclusion are disappointing. Major issues come from lack of markers for subpopulations, unavailability of codes, unavailability of parameters in cell clustering and population identification, failed integration of scRNA-seq/scTCR-seq data, and incomplete or misapplied marker sets for CyTOF subpopulations. These weaknesses require to be addressed substantially; otherwise, the novelty with reproduction and rigor of the research would be significantly reduced.

1. To avoid manual annotation of cell populations, authors adopted Azimuth function in Seurat to annotate cell subtypes. Thus, they bypassed one popular way to analyze scRNA-seq data by UMAP+cell markers+heatmap, which prevents the audience from further checking the defined cell populations. Generally, specific subpopulations may be more important for specific applications, requiring experience from immunologists to verify the used markers. One example is the “terminal exhausted T cell” redefined in the revision. These specific subpopulations are not easy for software to figure out; instead marker panel and associated heatmap or feature plot become essential to identify them.
2. Descriptions of scRNA-seq data and scTCR-seq data are piece by piece. scTCR-based TCR repertoire is a plus for scRNA-seq and thus their integration should be added to the analysis. Whether T cell usage and activation contribute to T cell phenotypes (functional populations) is still unknown. “We also searched for the CDR3 β amino acid sequences within our scRNAseq dataset to determine if the receptor was associated with a particular T cell phenotype. Due to the shallow depth of single-cell sequencing, this revealed only 4 CDR3 β amino acid sequences, limiting our ability to draw further conclusions.” This claim may be not true. Since scRNA-seq and scTCR-seq are paired, the cells sequenced by both platforms should be aligned by cell barcodes. So the above statement cannot be a reason for the lack of this analysis. The bottom line for this integration is to examine TCR activation and expression in the scRNA-seq data for the identified clones.
3. The population markers used in CyTOF are in problem. Markers shown in Sup Tab 2 or Sup Tab 3 are common for certain different populations. “CD4 central memory T cell” and “Naive T helper cells” are annotated by the same markers “CD3” and “CCR7.” In addition, these markers are insufficient for “CD4 central memory”; instead, CD4, CD45RO, and CD62L are still needed. The CyTOF data are the only dataset from tissue but the analysis is totally messy. Without source codes used in the analysis, it is unclear whether it is a problem caused by mistakes or just as it is.
4. Missing source codes and lacking a detailed description of each analysis in parameters in the applied software package. To allow the audience to repeat the results shown in the manuscript, it is required by “Code Availability” to include all source codes for data analyses and figures.
5. Current conclusion on the difference before and one year after lenalidomide exposure is mainly on “CD4 central memory”, “Naive T cells”, and “Naive B cells,” or no difference in TCR amino acid sequences. Sup Fig. 4 no statistical p values, how did the cell populations were selected? CD4:CD8 ratio is also in the conclusion but the data from Sup Fig. 5 does not support the conclusion (no statistical significance). These results might be biased by either PBMC sample or analyses. As pointed out by other reviewers, naïve T cells or B cells are frequently found in PBMC. Analyses shown in Fig. 2A-B are less powerful in which there is only a general UMAP with trajectory, and populations in Fig. 2C-E are too general. The right way to analyze the subpopulations for CD4, CD8 T cells, and B cells is to re-cluster of the cells from each type. A new UMAP is needed to annotate subpopulations. Unfortunately, no such results are shown in either main figure or sup figures. This is because CD4 T cells or CD8 T cells are more complex than we expected. Most of the time, they are polyfunctional, with markers from not only effector but exhaustion and memory. A simple marker combination or a general population is difficult to illuminate them.

Reviewer #1 (Remarks to the Author):

I would like to thank the authors for revising their manuscript and addressing the comments I raised. One last comment is that in the results section in the updated manuscript, the authors mentioned that "In general, patients with sustained MRD negativity were more likely to be younger, lack high-risk features such as ISS III disease or high-risk cytogenetics, and not have received high-dose melphalan with autologous stem cell transplant (HDM-ASCT)". I believe this sentence needs to be revised or removed as per Table 1 results, the only statistically significant difference is the High-risk cytogenetics. So this needs to be fixed to reflect the results as it gives the idea that these were significant differences. Otherwise, I believe the manuscript now is more clear and in better shape after acknowledging some of the limitations.

Thank you for this suggestion. We have revised the manuscript to list only the significant differences reported in table 1 (page 5).

Reviewer #2 (Remarks to the Author):

In this revised manuscript Coffey et al. profiled the immune system in patients with newly diagnosed MM receiving continuous lenalidomide maintenance therapy with the aim of uncovering correlates of long-term treatment response. The authors used single-cell RNA sequencing and T cell receptor β sequencing of the peripheral blood and CyTOF mass cytometry of the bone marrow to characterize the immune landscape in 23 patients before and one year after lenalidomide exposure. The authors compared patients achieving sustained minimal residual disease (MRD) negativity to patients who never achieved or were unable to maintain MRD negativity. While I believe that the analyses that the authors are reporting can support the future understanding of the importance of the immune environment in keeping the myeloma cancer cells under control. I still don't believe that the information contained in this manuscript can clearly answer specific questions, besides the collection of many cutting edge analyses in primary patients. If the authors could at least identify possible differences at baseline that can prospectively inform which group of patients can achieve long term MRD negativity versus others that would be important and increase the significance of their results. I still believe that the analysis at 1 year are not informative about the role of the immune system to maintain sustained minimal residual disease (MRD) negativity or to inform about the possible effect of lenalidomide. In my previous comments I asked if early response to lenalidomide including expansion of peripheral blood cytotoxic T cells could correlate with MRD negativity but this information has not been included in the revised form of the manuscript. This analysis could be also performed in a different set of patients. Overall this is a well written manuscript reporting very comprehensive correlative studies but the results are not particularly informative in the way they are reported.

We appreciate your comments and acknowledge that pre-treatment immunologic biomarker that predicts depth of response to lenalidomide maintenance therapy would benefit the field. In our study, we report numerous features that are associated with attaining sustained MRD negativity at baseline, defined as after induction but before initiation of lenalidomide maintenance. These include:

- Increased circulating naïve CD8⁺ T cells and CD56⁺ bright NK cells are observed at baseline in patients achieving sustained MRD negativity (Figure 2F).
- Decreased bone marrow CD4⁺ CD8⁺ T cells and plasmacytoid dendritic cells are observed at baseline in patients achieving sustained MRD negativity (Figure 4E).
- The presence of 74 circulating TCR β CDR3 amino acid sequences are specific to patients with sustained MRD negativity (Figure 5C).

- Numerous genes are found differentially expressed at baseline within T cells (Figure S6), B cells (Figure S8), and myeloid cells (Figure S10).

We did not find evidence that expansion of peripheral blood cytotoxic T cells associate with MRD status. While larger studies are needed to confirm these findings, our study does suggest there are immunologic features preceding immunotherapy that are associated with durable treatment outcomes. Once validated, we believe these findings have the potential to be used as a predictive biomarker of sustained MRD negativity prior to lenalidomide maintenance.

Reviewer #3 (Remarks to the Author):

In this report Coffey analyze with innovative techniques de immune systems and impact on MRD and its sustainability. The manuscript is of merit however the main limitation is the small number of cases studies, this could compromise some results.

In the discussion, we have acknowledged the small sample size may impact our conclusions and point out that a larger study is needed to verify our conclusions (page 15).

Reviewer #4 (Remarks to the Author):

Thank you for the detailed revision of the manuscript and for taken up several of my suggestions and performing additional analyses. While some new analyses are performed admirably and the resulting data contribute meaningfully to the manuscript (namely the TCR analysis), others have not done so. Several original comments have not been addressed convincingly.

Imbalance of patient groups

Thank you for including statistical information regarding the differences between the ASCT and no-ASCT group in Table 1. I could not find the “statement .. the discussion acknowledging that differences in disease biology and treatment history may have confounded our ability to reveal differences in the immune microenvironment”.

We sincerely appreciate your thorough feedback on our manuscript. We have acknowledged multiple limitations that may have confounded our ability to reveal differences in the immune microenvironment. This includes:

- **Heterogeneous treatment histories:** “To account for heterogeneity in up-front therapies, we studied patients with and without HDM-ASCT separately.” (Page 15)
- **Limited population size:** “Since this inherently resulted in comparison of a smaller number of patients, additional studies in a larger population are needed to confirm and expand on our findings.” (Page 15)
- **Inability to identify cause and effect relationships:** “Our observations do not inform on the possible function of lenalidomide, or whether these immune correlates are a consequence of low disease burden, allowing a return to a healthier immune system, or whether they indicate a contribution from immune microenvironment to disease control.” (Page 15)

Differential gene expression analysis of single cell RNA sequencing data: Page 7, the speculation about relevance of ERG1 should be relocated to the discussion. I would caution against such extensive speculation for the differential expression of a single gene, seemingly chosen at random. Do the authors genuinely believe the activation of central memory T cells

(defined by a single gene whose activity, namely induction of Tbet and STAT1, is not even shown) is significantly involved in MRD negativity? If so, how would the data support this?

We have removed this speculative statement from the manuscript and replaced it with a more general description of our observation of differentially expressed genes across MRD negative subgroups (Page 7).

Page 9, the sentence beginning “central memory T cells proliferate extensively and are...” is too interpretative for the results and should be moved to the discussion.

We observed CD4 central memory and effector memory T cells were the most abundant T cell types on average in patients with non-sustained MRD negativity and no prior transplant. To orient the reader who may be unfamiliar with the differences in central memory and effector memory T cells. We provide the following statement:

Central memory T cells proliferate extensively and are predominately located in the blood and secondary lymphoid tissues whereas effector memory T cells are less proliferative and are more commonly found in the spleen.(Mueller et al., 2013)

The intent of this statement is to be informative and not an interpretation of our results.

Definition of exhausted cells: Thank you for the additional analysis and the attempt to resolve exhausted cells. Unfortunately although the text describes exhaustion-associated genes, the actual expression of these genes is not shown. Furthermore, this classification of exhaustion is irregular. While the co-expression of GZMB, ID2 and BLIMP1 could describe exhausted cells, the selective expression of at least one of TIM3, CD244, CD39, CD160, CD200, LAG3, TIGIT, or CTLA4 casts a wide net regarding T cell phenotype. For instance, CD160 is expressed by NKT cells. Analysis of terminally exhausted cells by 10X scRNAseq will readily reveal cells with high-level co-expression of these checkpoint molecules, not only one. I note that TOX, a widely-used transcriptional marker of exhausted T cells (van der Leun, Thommen and Schumacher, 2020; Ren et al., 2021), was not significantly expressed in this dataset. Does this not suggest exhausted cells were not detected? And what of other single-cell RNAseq markers described in these reviews such as LAYN or CXCL13? Was the dataset at any point filtered to remove such genes? To be convinced that these genuinely resemble exhausted T cells, could the authors produce a per-cell heatmap (not a cluster-aggregated dotplot) and per-gene UMAPs showing the expression of the genes the authors used to classify exhausted cells (as well as LAYN and CXCL13), and show these cells regularly co-express these genes. To re-iterate, some checkpoint molecules are also activation markers. Perhaps best to use a description of terminal effectors simply expressing markers of activation.

The definition of T cell exhaustion by single-cell RNA sequencing varies across studies and it appears no two publications use the same criteria. For example, in Table 1 of the cited study from van der Leun *et al.* (*Nature Reviews Cancer* 2020), six separate gene signatures have been defined to describe exhausted T cells in human cancer which include:

- Guo *et al.* (NSCLC): LAYN, LAG3, TIGIT, PDCD1, HAVCR2, CTLA4, ITGAE
- Li *et al.* (melanoma): LAG3, PDCD1, CXCL13, TIGIT
- Sade-Feldman *et al.* (melanoma): LAG3, PDCD1, HAVCR2, ENTPD1, CTLA4
- Tirosh *et al.* (melanoma): PDCD1, HAVCR2, TIGIT, LAG3, CTLA4
- Yost *et al.* (BCC): LAG3, HAVCR2, PDCD1, GZMB, ENTPD1, ITGAE
- Zhang *et al.* (CRC): LAYN, HAVCR2, CXCL13, PDCD1, IFNG, ITGAE

- Zheng *et al.* (HCC): *LAYN, PDCD1, HAVCR2, CTLA4*

Furthermore, it has not been determined if expression of any, all, or a subset of these genes are necessary for a T cell to become exhausted. While our original definition was derived from genes that distinguish terminally exhausted from precursor exhausted T cells (Kallies *et al.*, 2020), we acknowledge that our signature has not been validated using single-cell sequencing. Therefore, we have chosen to take a more pragmatic approach and show the percentage of T cells expressing the previously described genes associated with T cell exhaustion in human cancers (revised Figure 3). We have updated our results (page 7) section to reflect this change. Additionally, we have created two new supplementary figures (revised Supplementary Figure 12 and 13) showing the expression of these genes within individual cells in the form of a heatmap and UMAP plot. You will note that although *TOX* gene expression was not present in our prior, multi-gene definition of exhausted T cells, its expression is detectable among T cells within our dataset.

However, ignoring the question of identity, the differential abundance or expression analysis does not seem to support the relevance of these cells in this cohort; the differences in Fig.3C lack significance and the differential expression within CD8+ Terminal Effectors (within which terminally exhausted CD8 cells sit) in Fig.S7 do not reveal exhaustion markers. Given these problems with definition I am uncomfortable with inclusion of this subset in Fig.6E.

Based on our revised analysis, we find evidence to suggest that T cells expressing the exhaustion marker *TIGIT* are more abundant in patients with non-sustained MRD negativity who do not undergo a transplant. Rather than referring to these as “exhausted T cells”, we have elected to call them *TIGIT*+ T cells throughout the manuscript and in Figure 6E. You will note that *TIGIT* is not a differentially expressed gene in Figure S7, but *TIGIT* expressing T cells are differentially abundant.

Figure 6: Comparison of immune cell frequencies in healthy donor and patient groups

The authors have not sufficiently taken into account the compositional heterogeneity amongst individual donors. Fig.6B-D appear to show correlations between donor-aggregated proportion for the different patient groups but this approach under-estimates the immune composition heterogeneity within each group. Could a compositional clustering approach similar to Combes *et al.*, (2022) be undertaken instead? Would such unsupervised compositional analysis reveal the same similarities between healthy and non-ASCT:Sus groups?

Combes *et al.* (*Cell* 2022) developed an unsupervised clustering technique to define “tumor immune archetypes” from bulk RNA sequencing. They validated their findings using bulk RNA sequencing on flow sorted immune populations. They demonstrated their method outperforms similar deconvolution algorithms (e.g. CIBERSORT).

Since we profiled our immune population at the single-cell level, there is no need to deconvolute our data into smaller units. Instead, we report the frequency of each immune cell type within each sample instead of assigning a sample to a broad category such as “T cell centric” or “myeloid centric” as was done in the referenced paper. We believe the additional granularity afforded by single-cell profiling enables greater insight into the immune microenvironment composition that would be missed by broad categorization into immune archetypes.

However, we do believe there is benefit to unsupervised clustering of samples as an orthogonal approach to confirm similarity of healthy and sustained MRD negative samples. Therefore, we

revised Figure 6B to include a heatmap and unsupervised hierarchical of cell frequency for each cell type averaged across subgroup (page 11). Samples are grouped according to K means clustering ($k = 2$). This demonstrates MGUS, SMM, and new diagnosed MM segregate from healthy and post treatment MM samples. This figure also more clearly shows the similarity between samples from patients who have and have not had a transplant as well as precursor and non-precursor disease states.

Finally, we removed progenitor cells from Figure 6A-6C since they are not differentiated immune cells.

TCR sequence analysis

Thank you for undertaking additional analysis and the interesting finding that several CDR3 beta are unique to patients with sustained MRD negativity is convincing and represents a promising avenue of investigation in the manuscript. Could the authors follow up this finding in more detail? For instance, what is the phenotype of these clones in the scRNAseq dataset? Do they share structural features in the CDR3 region? Could these clones be used to derive a gene signature of prognostic relevance? I appreciate this may not be possible but could these sequences be generated in vitro and their reactivity assessed against autologous tumour material, similar to Caushi et al., (2021).

The phenotype of the differentially abundant TCR β CDR3 sequences shown in figure 5C are listed in Supplemental table 3. While we agree that determining the antigen specificity of the 74 differentially abundant TCR β is of great interest, doing so would be a substantial challenge since we do not know the pairing CDR3 α sequence for the majority of receptors as they were not detectable within the single-cell RNAseq data. However, we did perform an analysis using GLIPH (grouping of lymphocyte interactions by paratope hotspots) (Glanville et al., 2017) to uncover shared TCR β CDR3 amino acid sequence motif suggesting an increased likelihood of shared antigen specificity among some of the receptors (Supplemental Table 3, Supplemental Figure 16).

Discussion

This is improved however suggest to focus on the no-ASCT cohort as here the correlations of immune cell frequencies with sustained MRD appear more consistent, eg. naïve CD4 and CD4CM cells, while the effect of ASCT is so dominant that it is more difficult, at least within the analyses conducted so far, to discern a consistent narrative around sustained MRD response. Suggest also to discuss the clinical implications of the unique shared CDR3 clones amongst patients with sustained MRD negative response. Suggest also that unless able to more convincingly resolve exhausted cells, to tone down the reference to this subset, and alter the narrative in Fig 6E.

Suggest remove the sentence “HDM-ASCT exerts a significant, long-term effect on T cell phenotype and does not appear to alter the ability of patients to achieve a durable remission.” as our extensive clinical experience would suggest otherwise. Instead re-phrase to say something like “..the changes due to ASCT preclude any consistent finding regarding correlation of T cell phenotype with sustained MRD response within the current analysis”.

We have replaced the term “exhausted T cells” with “TIGIT+ T cells” since that more clearly describes the identified cell population. We have also replaced the phrase “HDM-ASCT exerts a significant, long-term effect on T cell phenotype and does not appear to alter the ability of patients to achieve a durable remission” with the recommended phrase “the changes due to

HDM-ASCT preclude any consistent finding regarding correlation of T cell phenotype with sustained MRD response within the current analysis.” (Page 12)

References:

Caushi, J. X. et al. (2021) ‘Transcriptional programs of neoantigen-specific TIL in anti-PD-1-treated lung cancers’, *Nature* 2021 596:7870, 596(7870), pp. 126–132. doi: 10.1038/s41586-021-03752-4.

Combes, A. J. et al. (2022) ‘Discovering dominant tumor immune archetypes in a pan-cancer census’, *Cell*, 185(1), pp. 184-203.e19. doi: 10.1016/J.CELL.2021.12.004.

van der Leun, A. M., Thommen, D. S. and Schumacher, T. N. (2020) ‘CD8+ T cell states in human cancer: insights from single-cell analysis’, *Nature Reviews Cancer. Nature Research*, pp. 218–232. doi: 10.1038/s41568-019-0235-4.

Ren, X. et al. (2021) ‘Insights Gained from Single-Cell Analysis of Immune Cells in the Tumor Microenvironment’, <https://doi.org/10.1146/annurev-immunol-110519-071134>, 39, pp. 583–609. doi: 10.1146/ANNUREV-IMMUNOL-110519-071134.

Reviewer #5 (Remarks to the Author):

This work studied sustained MRD negativity in patients with multiple myeloma using 3 single-cell techniques to longitudinally characterized the immune landscape in 23 patients before and one year after lenalidomide exposure. Study design, hypothesis generation, and clinical samples are great; however, data analysis and conclusion are disappointing. Major issues come from lack of markers for subpopulations, unavailability of codes, unavailability of parameters in cell clustering and population identification, failed integration of scRNA-seq/scTCR-seq data, and incomplete or misapplied marker sets for CyTOF subpopulations. These weaknesses require to be addressed substantially; otherwise, the novelty with reproduction and rigor of the research would be significantly reduced.

1. To avoid manual annotation of cell populations, authors adopted Azimuth function in Seurat to annotate cell subtypes. Thus, they bypassed one popular way to analyze scRNA-seq data by UMAP+cell markers+heatmap, which prevents the audience from further checking the defined cell populations. Generally, specific subpopulations may be more important for specific applications, requiring experience from immunologists to verify the used markers. One example is the “terminal exhausted T cell” redefined in the revision. These specific subpopulations are not easy for software to figure out; instead marker panel and associated heatmap or feature plot become essential to identify them.

Automated cell classification reduces bias and improves reproducibility. Supplemental Figure 2 shows the top genes most specifically expressed per cell type along with the log(mean + 1) expression and proportion of cells expressing that marker. This figure shows the expected expression of genes across automated cell types. For example, *CD79A* is expressed by B cell, *CD8A* is expressed by CD8 T cells, and *FOXP3* is expressed by regulatory T cells.

Since we mapped our dataset to a reference PBMC dataset that does not contain dysfunctional T cells typically present within the tumor microenvironment, we also employed manual classification using a gene list specific to exhausted T cells. To clarify the expression of these genes, we have added an additional heatmap and several UMAP plots showing the expression of these genes at the single cell level (Supplemental Figures 12-13).

2. Descriptions of scRNA-seq data and scTCR-seq data are piece by piece. scTCR-based TCR repertoire is a plus for scRNA-seq and thus their integration should be added to the analysis.

Whether T cell usage and activation contribute to T cell phenotypes (functional populations) is still unknown. “We also searched for the CDR3 β amino acid sequences within our scRNAseq dataset to determine if the receptor was associated with a particular T cell phenotype. Due to the shallow depth of single-cell sequencing, this revealed only 4 CDR3 β amino acid sequences, limiting our ability to draw further conclusions.” This claim may be not true. Since scRNA-seq and scTCR-seq are paired, the cells sequenced by both platforms should be aligned by cell barcodes. So the above statement cannot be a reason for the lack of this analysis. The bottom line for this integration is to examine TCR activation and expression in the scRNA-seq data for the identified clones.

T cell receptor sequencing of the blood was performed using two separate technologies. To capture more CDR3 β sequences per sample, we used the ImmuneSeq platform (referred to as TCR β sequencing in the manuscript). To capture paired CDR3 α and CDR3 β sequences along with gene expression we used scRNAseq. After excluding doublets (cells with both T and B cell receptors), unpaired CDR3 α and CDR3 β receptors, and receptors containing multiple different alpha or beta chains, we found that 50% of cells captured by scRNAseq also had pairing gene expression (Supplemental Figure 14).

Using the ImmuneSeq platform, we detected 74 differentially abundant CDR3 β sequences specific to patients achieving sustained MRD negativity. To determine the phenotype of these T cell clones, we searched for the CDR3 β sequences within the scRNAseq data performed on the same blood sample, but only detected 4 CDR3 β sequences in common with both datasets (Supplemental Table 3, column C). This was not surprising since scRNAseq has a shallow depth of scRNAseq, which captured only 18,382 unique receptor sequences (Supplemental Figure 14), compared to TCR β sequencing, which captured 1,311,455 unique and CDR3 β sequences.

3. The population markers used in CyTOF are in problem. Markers shown in Sup Tab 2 or Sup Tab 3 are common for certain different populations. “CD4 central memory T cell” and “Naive T helper cells” are annotated by the same markers “CD3” and “CCR7.” In addition, these markers are insufficient for “CD4 central memory”; instead, CD4, CD45RO, and CD62L are still needed. The CyTOF data are the only dataset from tissue but the analysis is totally messy. Without source codes used in the analysis, it is unclear whether it is a problem caused by mistakes or just as it is.

We used Astrolabe analysis platform (<https://astrolabediagnosics.com>) to automatically identify cell subsets based on our antibody panel. We have revised the methods section to provide additional details regarding the labeling algorithm used by Astrolabe (page 18). We have also added more details to supplemental table 2 to clarify the cell markers. The table should be interpreted as a hierarchy. For example, CCR7+ CD4 central memory T cells are a subset of CD4+ T cells (CD45RA-), which are a subset of CD4+ T cells, which are a subset of CD3+ CD19- CD56- CD33- T cells. In contrast, CCR7+ Naive T helper cells are a subset of CD4+ T Cells (CD45RA+), which are a subset of CD4+ T cells, which are a subset of CD3+ CD19- CD56- CD33- T cells. Therefore, CD4 central memory T cells are distinguishable from Naive T helper cells on the basis of expression of CD45RA (Finak et al., 2016; Maecker et al., 2012)

4. Missing source codes and lacking a detailed description of each analysis in parameters in the applied software package. To allow the audience to repeat the results shown in the manuscript, it is required by “Code Availability” to include all source codes for data analyses and figures.

We had added an additional section to the supplemental methods detailing the key R functions and related parameters used for our data analyses. Additionally, to improve reproducibility, we are now including cell counts for our scRNAseq and CyTOF datasets as supplemental tables 4 and 5.

5. Current conclusion on the difference before and one year after lenalidomide exposure is mainly on “CD4 central memory”, “Naïve T cells”, and “Naïve B cells,” or no difference in TCR amino acid sequences. Sup Fig. 4 no statistical p values, how did the cell populations were selected? CD4:CD8 ratio is also in the conclusion but the data from Sup Fig. 5 does not support the conclusion (no statistical significance). These results might be biased by either PBMC sample or analyses. As pointed out by other reviewers, naïve T cells or B cells are frequently found in PBMC. Analyses shown in Fig. 2A-B are less powerful in which there is only a general UMAP with trajectory, and populations in Fig. 2C-E are too general. The right way to analyze the subpopulations for CD4, CD8 T cells, and B cells is to re-cluster of the cells from each type. A new UMAP is needed to annotate subpopulations. Unfortunately, no such results are shown in either main figure or sup figures. This is because CD4 T cells or CD8 T cells are more complex than we expected. Most of the time, they are polyfunctional, with markers from not only effector but exhaustion and memory. A simple marker combination or a general population is difficult to illuminate them.

Throughout the manuscript, statistical significance and fold change are shown in volcano plots of differentially abundant cells and expressed genes. Supplemental figures 6-11 show individual UMAPs of T cells, B cells, and myeloid cell sub-clusters color coded by the samples' MRD negative status.

REFERENCES

- Finak, G., Langweiler, M., Jaimes, M., Malek, M., Taghiyar, J., Korin, Y., Raddassi, K., Devine, L., Obermoser, G., Pekalski, M. L., Pontikos, N., Diaz, A., Heck, S., Villanova, F., Terrazzini, N., Kern, F., Qian, Y., Stanton, R., Wang, K., ... McCoy, J. P. (2016). Standardizing Flow Cytometry Immunophenotyping Analysis from the Human ImmunoPhenotyping Consortium. *Scientific Reports*, 6(1), 20686. <https://doi.org/10.1038/srep20686>
- Glanville, J., Huang, H., Nau, A., Hatton, O., Wagar, L. E., Rubelt, F., Ji, X., Han, A., Krams, S. M., Pettus, C., Haas, N., Arlehamn, C. S. L., Sette, A., Boyd, S. D., Scriba, T. J., Martinez, O. M., & Davis, M. M. (2017). Identifying specificity groups in the T cell receptor repertoire. *Nature*, 547(7661), 94–98. <https://doi.org/10.1038/nature22976>
- Kallies, A., Zehn, D., & Utzschneider, D. T. (2020). Precursor exhausted T cells: key to successful immunotherapy? *Nature Reviews Immunology*, 20(2), 128–136. <https://doi.org/10.1038/s41577-019-0223-7>
- Maecker, H. T., McCoy, J. P., & Nussenblatt, R. (2012). Standardizing immunophenotyping for the Human Immunology Project. *Nature Reviews Immunology*, 12(3), 191–200. <https://doi.org/10.1038/nri3158>

Mueller, S. N., Gebhardt, T., Carbone, F. R., & Heath, W. R. (2013). Memory T Cell Subsets, Migration Patterns, and Tissue Residence. *Annual Review of Immunology*, 31(1), 137–161. <https://doi.org/10.1146/annurev-immunol-032712-095954>

REVIEWER COMMENTS

Reviewer #2 (Remarks to the Author):

First of all, I thank you the authors to address my previous concerns about possible baseline biomarkers correlated with achieving long term MRD negativity.

In my point of view if this manuscript was focused on the identification of specific biomarkers to further characterize MRD, then a patient validation cohort should be needed. If the authors are focusing on the biological role of the immune environment in maintaining sustained MRD then functional experiments should be also conducted. The authors themselves admit in several instances that these findings need to be validated in a larger cohort of patients, but at the same time they are trying to publish no yet validated scientific conclusions.

This is an observational study with few interesting preliminary data that need to be validated in a bigger cohort of patients.

Reviewer #4 (Remarks to the Author):

Imbalance of patient groups

Thank you for your edits describing the limitations imposed by sample size and patient heterogeneity and for acknowledging that the correlates shown in your analysis do not imply any function of lenalidomide, as causal relationships cannot be inferred.

Differential gene expression of single cell RNA sequencing data

Thank you for moving speculation of EGR1 to the discussion.

Definition and identification of exhausted cells

Thank you for edits on this topic, and we agree that the definition of exhausted cells varies across studies. I agree with displaying the data as the percentage of T cells expressing different markers in Fig 3A and the associated heatmap and per-gene UMAPs in Suppl Fig 12, 13 makes more sense and is clearer. In this context, I am unsure that any conclusion can be drawn about exhausted T cells.

Confining your description to TIGIT-expressing cells is reasonable, but expressed as % of all T cells does not impart much biological insight. May I suggest that you confine your analysis of TIGIT+ve cells to an effector subpopulation, eg. PD1+ TEM? As otherwise the % of T cells expressing a given marker is likely a result of differences in composition (i.e. low % TIGIT+ may simply mean high % of TIGIT-negative naïve cells). The y axis could be split, or show low-percentage genes shown on a different plot. Additionally, the jittered points are overlapping between the two groups and should be replaced with `geom_point``.

Ignoring this, if the finding of 3A and B are true this may be interesting, though largely a negative finding (it appears most Tex markers and percentage of cells expressing them are unchanged, excluding TIGIT+ cells). This is a fair conclusion, but would lead us to agree that the exhaustion pathway is uninvolved here.

Composition of immune subsets

Thank you for the additional analyses.

I apologise for my lack of clarity in previous comments. My concerns regarding donor-aggregated cell type proportions are because this approach under-estimates the immune composition heterogeneity within each individual group. For instance, in the newly created Fig 6B, there is roughly 20% Plasmablasts in the MM group. But, individual donors may vary in the % plasmablasts (some under 20%, some higher). This analysis and associated plot fail to capture this heterogeneity.

I suggest that you create a plot similar to 4A, showing individual donor immune composition, with the patient groups shown as columnal colours. If this analysis of all cells is too noisy, it could be repeated on only T cells (abundance as a % T cells) or Myeloid cells. If using all bone marrow immune cells, I

would suggest removing plasmablasts and showing % of non-PC to ensure the presence of tumour does not skew immune compositional quantification. The hierarchical clustering of these matrices will be informative about the behaviour of individual patients with ASCT/no ASCT and Sus/non-sus MRD.

Reviewer #5 (Remarks to the Author):

This revision has addressed some questions on TCR β sequencing and markers for CyTOF analysis. After checking through the revised manuscript, there are still concerns about markers used for scRNA-seq, conclusions on TIGIT+ exhaustion cells, and data and code availabilities.

Majors:

1. "Automated cell classification reduces bias and improves reproducibility." It is true but insufficient for this work. One issue with this analysis is that it lacks the power to identify a specific subpopulation or subcluster for the phenotype of interest. Such a subpopulation or a subcluster should be identified by visualization and annotation of the cluster/subcluster using Seurat analysis. In your response to another reviewer and mine, you found TIGIT+ T cells. However, could you annotate this population in the scRNA-seq cluster/subcluster? Using reference populations in the PBMC is not reasonable because your study is specific and the shown subpopulation is also specific. From Fig. 2A to Fig.3A, there is a gap in the analysis for T cells. Either CD4 subgroup or CD8 subgroup should be further reclustered, and to find out which subcluster(s) is more unique. Based on the found subcluster (subpopulation), differential analysis can figure out which markers are more meaningful for explaining or annotating the specific subpopulation. As shown in Sup Fig 12 and Fig. 3A, critical molecular info was marked by the cells in the current annotation and TIGIT might not be the right marker to annotate the real specific subpopulation.

2. Data and code are still unavailable. (<https://crescent.cloud>, CRES-P31), no data there. Please upload your processed data to this cloud platform as well as consider GEO for publishing your data where more people search cancer scRNA-seq data. Your codes for scRNA-seq, CyTOF, and TCR analyses that were customized for your data should be uploaded into GitHub, with markdown documents from R or Python notebooks for reviewers and audiences to repeat your analyses. Currently citing used tools in the "Code availability" is not OK.

Minors:

1. line 521: upplemental->supplemental.
2. line 208: Fig. 3E not exist.
3. line 194: Wilcoxin test?? Is it "Wilcoxon rank sum test"

REVIEWER COMMENTS

Reviewer #2 (Remarks to the Author):

First of all, I thank you the authors to address my previous concerns about possible baseline biomarkers correlated with achieving long term MRD negativity. In my point of view if this manuscript was focused on the identification of specific biomarkers to further characterize MRD, then a patient validation cohort should be needed. If the authors are focusing on the biological role of the immune environment in maintaining sustained MRD then functional experiments should be also conducted. The authors themselves admit in several instances that these findings need to be validated in a larger cohort of patients, but at the same time they are trying to publish no yet validated scientific conclusions.

This is an observational study with few interesting preliminary data that need to be validated in a bigger cohort of patients.

We greatly appreciate the reviewer's insightful comments and agree that further validation of our findings in a larger cohort of patients is necessary. Additionally, we acknowledge that performing functional experiments would be valuable to confirm our observations, but we are unable to do so due to resource limitations. Nonetheless, we firmly believe that our findings are novel and provide valuable insights into the biological mechanisms underlying sustained MRD. We have highlighted the limitations of our study in the discussion, and we believe that our results will stimulate future research in this area. We believe that future studies will build upon our findings and ultimately lead to new treatments approaches for MM.

Reviewer #4 (Remarks to the Author):

Imbalance of patient groups

Thank you for your edits describing the limitations imposed by sample size and patient heterogeneity and for acknowledging that the correlates shown in your analysis do not imply any function of lenalidomide, as causal relationships cannot be inferred.

Differential gene expression of single cell RNA sequencing data

Thank you for moving speculation of EGR1 to the discussion.

Definition and identification of exhausted cells

Thank you for edits on this topic, and we agree that the definition of exhausted cells varies across studies. I agree with displaying the data as the percentage of T cells expressing different markers in Fig 3A and the associated heatmap and per-gene UMAPs in Suppl Fig 12, 13 makes more sense and is clearer. In this context, I am unsure that any conclusion can be drawn about exhausted T cells.

Confining your description to TIGIT-expressing cells is reasonable, but expressed as % of all T cells does not impart much biological insight. May I suggest that you confine your analysis of TIGIT+ve cells to an effector subpopulation, eg. PD1+ TEM? As otherwise the % of T cells expressing a given marker is likely a result of differences in composition (i.e. low % TIGIT+ may simply mean high % of TIGIT-negative naïve cells). The y axis could be split, or show low-percentage genes shown on a different plot. Additionally, the jittered points are overlapping between the two groups and should be replaced with `geom_point`.

Thank you for this suggestion. We have revised Figure 3A so that the denominator is terminal effector memory T cells (T_{EM} , $n = 21,773$) rather than all T cells ($n = 59,451$). There were too

few PD1+ T_{EM} (n = 905) for a meaningful analysis. Since *CXCL13* and *LAYN* were expressed at nearly undetectable levels in this subpopulation (Figure S13), we removed these genes from figure 3A. We also changed the display of the points using the `geom_point` function as recommended. The resulting figure shows that T_{EM} cells expressing exhaustion genes are consistently more frequent in patients who did not achieve sustained MRD negativity. Within the T_{EM} subpopulation, we observed that while the difference in the abundance of TIGIT+ cells was no longer statistically significant, the frequency of TOX+ cells was significantly higher in patients who did not achieve sustained MRD negativity. This finding appeared most pronounced after one year of maintenance therapy, especially among patients who did not undergo a prior autologous stem cell transplant. These results support the role of autologous stem cell transplant in reducing T cell exhaustion and indicate the post-transplant period may be an optimal time to initiate immunotherapy.

Ignoring this, if the finding of 3A and B are true this may be interesting, though largely a negative finding (it appears most Tex markers and percentage of cells expressing them are unchanged, excluding TIGIT+ cells). This is a fair conclusion, but would lead us to agree that the exhaustion pathway is uninvolved here.

Since we were not able to consistently observe a statistical significance difference in T cell exhaustion genes between sustained and non-sustained MRD negativity, we have revised the text (page 8) and Figure 6 to de-emphasize this finding. Nevertheless, we do feel it is important to leave the analysis in the manuscript since our results show a trend toward a difference being present for most genes with statistical significance being reached for *TOX*, a crucial transcription factor involved in T cell exhaustion. We also acknowledge that future studies from larger populations are needed to confirm this finding.

Composition of immune subsets

Thank you for the additional analyses. I apologise for my lack of clarity in previous comments. My concerns regarding donor-aggregated cell type proportions are because this approach under-estimates the immune composition heterogeneity within each individual group. For instance, in the newly created Fig 6B, there is roughly 20% Plasmablasts in the MM group. But, individual donors may vary in the % plasmablasts (some under 20%, some higher). This analysis and associated plot fail to capture this heterogeneity. I suggest that you create a plot similar to 4A, showing individual donor immune composition, with the patient groups shown as columnal colours. If this analysis of all cells is too noisy, it could be repeated on only T cells (abundance as a % T cells) or Myeloid cells. If using all bone marrow immune cells, I would suggest removing plasmablasts and showing % of non-PC to ensure the presence of tumour does not skew immune compositional quantification. The hierarchical clustering of these matrices will be informative about the behaviour of individual patients with ASCT/no ASCT and Sus/non-sus MRD.

Thank you for the additional clarification. We did observe heterogeneity across individual samples from within disease subgroups. We have created a new figure using your suggestion to illustrate this (Figure S20). This heterogeneity was observed whether we clustered by all cell types or major cell groups (T cells, NK cells, B cell, myeloid cells, and dendritic cells). This heterogeneity is expected given the diversity in age, treatment history, and other clinical factors. We believe that by showing the data in aggregate reduces noise and makes it easier identify trends.

Whether we include plasmablasts in the analysis or not, the conclusions remain the same. That is, healthy donors share the greatest similarity to patients achieving sustained MRD negativity

without prior transplant. Likewise, MGUS, SMM, and MM samples cluster separately from healthy donors and treated patients using K means clustering. We have revised figures 6A-D to exclude plasmablasts from the dominator.

Reviewer #5 (Remarks to the Author):

This revision has addressed some questions on TCR β sequencing and markers for CyTOF analysis. After checking through the revised manuscript, there are still concerns about markers used for scRNA-seq, conclusions on TIGIT+ exhaustion cells, and data and code availabilities.

Majors:

1. "Automated cell classification reduces bias and improves reproducibility." It is true but insufficient for this work. One issue with this analysis is that it lacks the power to identify a specific subpopulation or subcluster for the phenotype of interest. Such a subpopulation or a subcluster should be identified by visualization and annotation of the cluster/subcluster using Seurat analysis. In your response to another reviewer and mine, you found TIGIT+ T cells. However, could you annotate this population in the scRNA-seq cluster/subcluster? Using reference populations in the PBMC is not reasonable because your study is specific and the shown subpopulation is also specific. From Fig. 2A to Fig.3A, there is a gap in the analysis for T cells. Either CD4 subgroup or CD8 subgroup should be further reclustered, and to find out which subcluster(s) is more unique. Based on the found subcluster (subpopulation), differential analysis can figure out which markers are more meaningful for explaining or annotating the specific subpopulation. As shown in Sup Fig 12 and Fig. 3A, critical molecular info was marked by the cells in the current annotation and TIGIT might not be the right marker to annotate the real specific subpopulation.

We appreciate your comments and acknowledge that there are multiple different approaches to cell classification using single-cell RNA sequence. Our approach combined automated cell classification to reduce bias and improve reproducibility as well as manual annotation to identify cell populations not represented in our reference dataset. For example, we labeled terminal effector memory T cells (T_{EM}) using an automated classifier and then identified T cells expressing exhaustion genes within the T_{EM} subgroup since exhausted T cells were not in our reference dataset. As you recommended, we annotated these T cells expressing exhaustion genes within the UMAP in Figure S13. Additionally, we re-clustered each cell subtype and performed a differential gene expression analysis between sustained and no-sustained MRD negative clusters as shown in Figures S6-11. We acknowledge that this analysis did not uncover genes associated with T cell exhaustion. However, our analysis was based on differential abundance between patient groups which may not correlate with differential expression (e.g. exhaustion genes may not be differentially expressed but exhausted T cells may be differentially abundant). Since our finding of differentially abundant exhausted T cells is only modestly significant, we have de-emphasized this finding in the revised manuscript (see revision on page 8 and revised figure 6).

2. Data and code are still unavailable. <https://crescent.cloud>, CRES-P31), no data there. Please upload your processed data to this cloud platform as well as consider GEO for publishing your data where more people search cancer scRNA-seq data. Your codes for scRNA-seq, CyTOF, and TCR analyses that were customized for your data should be uploaded into GitHub, with markdown documents from R or Python notebooks for reviewers and audiences to repeat your analyses. Currently citing used tools in the "Code availability" is not OK.

Our single-cell RNA sequencing data has been uploaded to CReSCENT. While the data is currently in a private archive, as a reviewer you may access the complete dataset data at <https://crescent.cloud> using this email address: coffey_2023@reviewers.crescent.cloud and password: coffey2023. Upon logging in, select the “Uploaded” tab at the top of the screen and you will find our dataset listed as “Multiple Myeloma (Coffey, Unpublished, 2023)”. Once the manuscript has been accepted, we will make the dataset freely available to the public. While we considered uploading our single-cell RNA sequencing data to GEO, our study consent does not allow us to share the raw sequencing fastq files through SRA or dbGAP which is a requirement of GEO. However, with CReSCENT, it is possible to download the original gene expression matrixes. To accomplish this, open our dataset, and select the download icon next to each sample. This will download a zipped file containing the barcodes.tsv.gz, features.tsv.gz, and matrix.mtx.gz files which can be imported into R using the Read10X function from the Seurat R package.

Our T cell receptor beta sequencing data has been uploaded to the ImmuneAccess database (<https://clients.adaptivebiotech.com/immuneaccess>, DOI 10.21417/DGC2023NC) and will be made publicly available upon acceptance of the manuscript. T and B cell receptor clonotypes from single-cell V(D)J sequencing are available in Supplementary Table 2. Single-cell barcodes are provided so that the data can be linked to the single-cell gene expression data on CReSCENT. CyTOF cell counts per sample are available in Supplementary Table 4.

We moved the R code from the supplemental materials to a GitHub repository which can be accessed from this website: <https://github.com/UM-Myeloma-Genomics/Immunophenotypic-correlates-of-sustained-MRD-negativity>. The code is heavily commented and shows step-by-step how single-cell RNA sequencing, single-cell V(D)J sequencing, CyTOF, and T cell receptor beta sequencing datasets were processed from their original form and analyzed for differential abundance.

Minors:

1. line 521: upplemental->supplemental. Corrected
2. line 208: Fig. 3E not exist. Corrected
3. line 194: Wilcoxin test?? Is it "Wilcoxon rank sum test" Corrected

REVIEWER COMMENTS

Reviewer #4 (Remarks to the Author):

Thank you to the authors for your further work on the definition and identification of exhausted cells. This is good now.

Composition of immune cells

Thank you to the authors for undertaking further analysis, now shown in new Fig S20. Can you please show the data to support the statement "This heterogeneity was observed whether we clustered by all cell types or major cell groups (T cells, NK cells, B cell, myeloid cells, and dendritic cells)" as it looks from Fig S20 that the majority of the cells from HD are classical monocytes.

Please can you extend fig 6D to show the abundance correlation between HD, and each of the following groups: No ASCT:Non-sus MRD-, ASCT:Sus MRD-, and ASCT:Non-sus MRD-. The results of these further correlative analyses are referred to in the Discussion, however it is important to display them as this is a key conclusion of the manuscript.

A further comment regarding the TCR β receptor analysis of blood (I appreciate that this was not highlighted in my previous review).

The authors state they have "detected highly specific TCR β sequences that were unique to the repertoire of patients with sustained MRD negativity". Can the authors demonstrate that these sequences are not the result of low level CDR3 sharing amongst different individuals, and not connected to MRD negativity? If the authors take each repertoire used for 5C, shuffle the sustained vs non-sustained labels for each individual patient, re-calculate overlap, and permute this 100 times, is the number of shared TCR β counts within the real groups higher vs the random shuffled label? If the real labels have a higher level of sharing than the shuffled labels, it is likely not simply random sharing between individuals. If they do not, then we should question the relevance of these TCRs.

Additionally, TCR sharing is greater between individuals with the same HLA genotype (see 10.1371/journal.pone.0249484). Can the authors use a chi-squared test to show the number of overlapping CDR3 between sustain MRD patients is not simply due to a shared HLA genotype?

Have the 14 CDR3 β amino acid sequences with known antigen specificity to microbial antigens and one human antigen been included in 5C? If so, they should be removed and this noted in the results, as shared microbial sequences in sustained MRD patients would preclude the necessity to characterise their antigen specificity.

The authors carried out LymphoSeqDB search and OLGA analysis, and conclude that "Together, these analyses revealed that T cell receptor β sequences that are unique to patients with sustained MRD negativity are infrequently found in healthy individuals and have a relatively low CDR3 sequence generation probability". Can the authors plot the results of this LymphoSeqDB search and OLGA analysis in figure 5? As these will directly affect the interpretation of 5C.

The value of parallel single cell sequencing is to be able to capture the phenotype of putative specific TCRs, however, the authors state that due to the shallow depth of single-cell sequencing, this revealed only 4 CDR3 β amino acid sequences, limiting our ability to draw in further conclusions." Despite the limited number of matches, the phenotype of these CDR3s is of interest and should be shown. A heatmap showing the number of cells of each CDR3 in the different T cell phenotypes should be shown.

The abundance information in 5C (% TRB) cannot be interpreted directly as these percentages are not placed in the context of TCR abundance in the full repertoire. Can the authors show, for three representative donors in 5C, the boxplot of overall abundance of all CDR3s in each sample, and plot individually as red jittered points (for geom_jitter, ensure "height=0") the abundance of the specific

CDR3s shown in 5C, so their abundance is contextualised in the full repertoire. The we can see if these sequences are notably larger, smaller or similar to the average abundance, and draw conclusions about their role in T cell immunity.

Can the authors show the abundance of the sequences in 5C at baseline and follow-up. Dynamics in abundance might help elucidate what role they play in sustained MRD negativity.

Discussion

Please consider replacing the statement “we uncovered distinct differences in immune cell composition specific to patients sustained MRD negativity that vary according to exposure to high-dose melphalan.” With a more nuanced statement “ we report particular features of immune cell composition present in patients with sustained MRD negativity that vary according to exposure to high-dose melphalan”.

And the statement “Our findings support the hypothesis that the immune microenvironment influences duration of treatment response..” with “Our findings support the hypothesis that the immune microenvironment reflects the duration of treatment response...”

Reviewer #5 (Remarks to the Author):

My questions and data availability issues were addressed.

REVIEWER COMMENTS

Reviewer #4 (Remarks to the Author):

Thank you to the authors for undertaking further analysis, now shown in new Fig S20. Can you please show the data to support the statement “This heterogeneity was observed whether we clustered by all cell types or major cell groups (T cells, NK cells, B cell, myeloid cells, and dendritic cells)” as it looks from Fig S20 that the majority of the cells from HD are classical monocytes.

We have added a second version to Figure S20 in the revised manuscript showing major cell types (T cells, NK cells, B cell, myeloid cells, and dendritic cells) that has been reproduce below. Both figures reveal imperfect clustering of disease groups based on frequency of immune cells. For example, while most healthy donors have high levels of classical monocytes, there is one MM sample that also has elevated monocytes (GSM3528777) while the other MM samples do not. Despite the heterogeneity within clusters, it is apparent from these figures that healthy donors and patients with no ASCT and sustained MRD negativity tend to co-localize in the same cluster (i.e. share a major branch of the dendrogram).

Please can you extend fig 6D to show the abundance correlation between HD, and each of the following groups: No ASCT:Non-sus MRD-, ASCT:Sus MRD-, and ASCT:Non-sus MRD-. The results of these further correlative analyses are referred to in the Discussion, however it is important to display them as this is a key conclusion of the manuscript.

We have added these requested figures to Figure S22.

A further comment regarding the TCR β receptor analysis of blood (I appreciate that this was not highlighted in my previous review). The authors state they have “detected highly specific TCR β sequences that were unique to the repertoire of patients with sustained MRD negativity”. Can the authors demonstrate that these sequences are not the result of low level CDR3 sharing amongst different individuals, and not connected to MRD negativity? If the authors take each repertoire used for 5C, shuffle the sustained vs non-sustained labels for each individual patient, re-calculate overlap, and permute this 100 times, is the number of shared TCR β counts within the real groups higher vs the random shuffled label? If the real labels have a higher level of sharing than the shuffled labels, it is likely not simply random sharing between individuals. If they do not, then we should question the relevance of these TCRs.

Compared to random chance, sharing of CDR3 β sequences was significantly less common among patients with sustained MRD negativity. We randomly shuffled the sustained and non-sustained labels for each patient, computed the number of CDR3 β amino acid sequences shared by 2 or more individuals with sustained MRD negativity, and permuted this 1,000 times. The histogram below shows the results of the permutation with the red line indicating the actual number of shared sequences by 2 or more individuals with sustained MRD negativity. These results show the number of shared sequences was less than all permuted values indicating the null hypothesis that shared CDR3 β sequences between the two groups are due to random chance should be rejected (P value = 0). This observation supports our findings reported in figure 5C, that the identified sequences unique to patients with sustained MRD negativity are unlikely to be a chance event.

Additionally, TCR sharing is greater between individuals with the same HLA genotype (see [10.1371/journal.pone.0249484](https://doi.org/10.1371/journal.pone.0249484)). Can the authors use a chi-squared test to show the number of overlapping CDR3 between sustain MRD patients is not simply due to a shared HLA genotype?

We did not find a particular HLA type to be more prevalent in patients with sustained MRD negativity compared to those without sustained MRD negativity. For this analysis, we chose to use Fisher's exact test instead of the chi-squared test due to the small sample size. For individuals in whom we performed TCR β sequencing, the frequency of each HLA type was not

significantly different between sustained and not sustained MRD negative patients. These results are also shown in Figure S18 in the revised manuscript.

Have the 14 CDR3 β amino acid sequences with known antigen specificity to microbial antigens and one human antigen been included in 5C? If so, they should be removed and this noted in

the results, as shared microbial sequences in sustained MRD patients would preclude the necessity to characterise their antigen specificity.

Yes, the 14 CDR3 β amino acid sequences with known antigen specificity to microbial antigens are included in Figure 5C and listed in Supplemental Table 3. Since a single TCR can potentially recognize more than 10^6 different MHC-bound peptides (J Biol Chem 6;287(2):1168-77, 2012), this does not preclude it from reacting to a non-microbial antigens. In fact, others have reported that virus-specific memory T cells can extend their surveillance to neoantigens expressed by the tumor cells (Nat Comm 10;567, 2019). For these reasons, we believe the 14 sequences should not be removed from the Figure 5C.

The authors carried out LymphoSeqDB search and OLGA analysis, and conclude that “Together, these analyses revealed that T cell receptor β sequences that are unique to patients with sustained MRD negativity are infrequently found in healthy individuals and have a relatively low CDR3 sequence generation probability”. Can the authors plot the results of this LymphoSeqDB search and OLGA analysis in figure 5? As these will directly affect the interpretation of 5C.

We have added the CDR3 β prevalence and generation probabilities to Figure 5C in the revised manuscript. The same information is also available in Supplemental Table 3.

The value of parallel single cell sequencing is to be able to capture the phenotype of putative specific TCRs, however, the authors state that due to the shallow depth of single-cell sequencing, this revealed only 4 CDR3 β amino acid sequences, limiting our ability to draw in further conclusions.” Despite the limited number of matches, the phenotype of these CDR3s is of interest and should be shown. A heatmap showing the number of cells of each CDR3 in the different T cell phenotypes should be shown.

The 4 relevant CDR3 β amino acid sequences detected by both ImmunoSeq and scRNAseq were found in 8 cells (1 clonotype was found in 5 cells and the remaining 3 clonotypes were found in 1 cell each). This information is provided in Supplemental Tables 2 and 3.

The abundance information in 5C (% TRB) cannot be interpreted directly as these percentages are not placed in the context of TCR abundance in the full repertoire. Can the authors show, for three representative donors in 5C, the boxplot of overall abundance of all CDR3s in each sample, and plot individually as red jittered points (for geom_jitter, ensure "height=0") the abundance of the specific CDR3s shown in 5C, so their abundance is contextualised in the full repertoire. The we can see if these sequences are notably larger, smaller or similar to the average abundance, and draw conclusions about their role in T cell immunity.

Supplemental Table 3 provides the observed, minimum, maximum, and mean frequency for each of the CDR3 β listed in Figure 5C so their abundance can be contextualized in the full repertoire.

Can the authors show the abundance of the sequences in 5C at baseline and follow-up. Dynamics in abundance might help elucidate what role they play in sustained MRD negativity.

For each sample in Figure 5C, the CDR3 β sequence abundance is shown for both baseline and follow-up time points.

Discussion

Please consider replacing the statement “we uncovered distinct differences in immune cell composition specific to patients sustained MRD negativity that vary according to exposure to high-dose melphalan.” With a more nuanced statement “ we report particular features of immune cell composition present in patients with sustained MRD negativity that vary according to exposure to high-dose melphalan”.

And the statement “Our findings support the hypothesis that the immune microenvironment influences duration of treatment response..” with “Our findings support the hypothesis that the immune microenvironment reflects the duration of treatment response...”

We thank you for your suggestions. Our preference is to keep the statements in the original form.

REVIEWERS' COMMENTS

Reviewer #4 (Remarks to the Author):

Compositional analyses

I thank the authors for the new Figure S20, showing clustering of subjects according to frequency of immune cell types, but I fail to see that healthy donors and subjects with no ASCT and sustained MRD tend to co-localize in the same cluster (i.e. share a major branch of the dendrogram). This is not the case in Fig 6B (as stated in the paper), nor in Fig S20A, where only one donor with No ASCT:SusMRD- (56-Follow-up) is in a cluster with several healthy. From this heatmap it seems this is mostly driven by the abundance of classical monocytes. The other No ASCT:SusMRD- samples seem fairly inter-mixed with other diagnoses. "Trend" also suggests statistics was performed, can you show these?

I thank the authors for including Fig.S22. From these plots it can be seen that the No ASCT:Sus MRD- patients do have a greater correlation in cell type abundance with healthy donors versus the other groups presented. However, it would appear all comparisons are positively correlated with healthy donor abundance, and it is not clear that the R value of 0.70 (and not 0.85 as stated in the text) is materially different from the other R values of 0.53 and 0.43. I think it would be reasonable to say that for all groups of patients, there was a strong positive correlation with healthy donors, but appeared stronger for No ASCT:Sus MRD-. I am uneasy about any stronger wording for these results.

TCR overlap

I thank the authors for undertaking the analysis of TCRB overlap between shuffled patient labels. By calculating TCRB overlap between 2 or more patients with shuffled sustained and non-sustained labels, the authors have created a background level of TCRB sharing to be expected in a group of individuals the same size as the non-sustained group (n=4). The histogram shows that, on average, roughly 24,750 shared sequences are to be expected between 2 or more patients in a group of this size.

The overlap between the sustained MRD- patients however was lower than this background overlap. If the patients with sustained MRD- have a higher level of sharing than the shuffled labels, this overlap within the real group (sustained MRD-) is likely not simply random sharing between individuals. As the real labels have less sharing than the background, it suggests TCRB overlap is less than expected by chance. Therefore, there is no enrichment of TCRB sharing within the sustained MRD- group. If the level of overlap was higher in the real data than background, it would instead suggest an enrichment for shared sequences and be of interest.

This analysis therefore argues against including Fig.5C, as this degree of TCRB overlap could be observed between any 4 independent individuals, independent of their sustained MRD- status.

Patient HLA background

The authors show that sustained MRD- patients do not overlap in their HLA genotype. This does support the hypothesis that HLA background alone is not contributing their TCR overlap – which was my original comment. However, if the sustained MRD- group do not share HLA genotypes, then any shared TCRs would be unlikely to have the same antigen specificity given they would be seeing antigen presented on (structurally distinct) HLA molecules. This TCRB overlap analysis therefore is simply reporting a level of overlap to be expected in a random group of patients of mixed HLA background.

REVIEWER COMMENTS

Reviewer #4 (Remarks to the Author):

Compositional analyses

I thank the authors for the new Figure S20, showing clustering of subjects according to frequency of immune cell types, but I fail to see that healthy donors and subjects with no ASCT and sustained MRD tend to co-localize in the same cluster (i.e. share a major branch of the dendrogram). This is not the case in Fig 6B (as stated in the paper), nor in Fig S20A, where only one donor with No ASCT:SusMRD- (56-Follow-up) is in a cluster with several healthy. From this heatmap it seems this is mostly driven by the abundance of classical monocytes. The other No ASCT:SusMRD- samples seem fairly inter-mixed with other diagnoses. "Trend" also suggests statistics was performed, can you show these?

We updated the hierarchical clustering method in Fig. S20 (Fig S19 in the revised manuscript) to Ward's minimum variance method. This method seeks to minimize the total within-cluster variance. Previously, we used the complex linkage method which considers the maximum distance between clusters for merging. However, this approach can be sensitive to outliers since a single outlier can increase the maximum distance. With Ward's minimum variance method, it can more clearly be seen that the No ASCT:SusMRD- samples share the same major branch of the dendrogram in Fig. S19A as the samples from healthy donors. Specifically, 4 of 8 ASCT:SusMRD- samples cluster with 8 of 9 healthy samples (Fisher P value 0.131).

I thank the authors for including Fig.S22. From these plots it can be seen that the No ASCT:Sus MRD- patients do have a greater correlation in cell type abundance with healthy donors versus the other groups presented. However, it would appear all comparisons are positively correlated with healthy donor abundance, and it is not clear that the R value of 0.70 (and not 0.85 as stated in the text) is materially different from the other R values of 0.53 and 0.43. I think it would be reasonable to say that for all groups of patients, there was a strong positive correlation with healthy donors, but appeared stronger for No ASCT:Sus MRD-. I am uneasy about any stronger wording for these results.

We have revised the manuscript to acknowledge that "a positive correlation between healthy bone marrow and all subgroups, but the strongest correlation was between healthy donors and patients achieving sustained MRD negativity and no prior HDM-ASCT" (Page 12, final sentence of the results section).

TCR overlap

I thank the authors for undertaking the analysis of TCRB overlap between shuffled patient labels. By calculating TCRB overlap between 2 or more patients with shuffled sustained and non-sustained labels, the authors have created a background level of TCRB sharing to be expected in a group of individuals the same size as the non-sustained group (n=4). The histogram shows that, on average, roughly 24,750 shared sequences are to be expected between 2 or more patients in a group of this size.

The overlap between the sustained MRD- patients however was lower than this background overlap. If the patients with sustained MRD- have a higher level of sharing than the shuffled labels, this overlap within the real group

(sustained MRD-) is likely not simply random sharing between individuals. As the real labels have less sharing than the background, it suggests TCRB overlap is less than expected by chance. Therefore, there is no enrichment of TCRB sharing within the sustain MRD- group. If the level of overlap was higher in the real data than background, it would instead suggest an enriching for shared sequences and be of interesting.

This analysis therefore argues against including Fig.5C, as this degree of TCRB overlap could be observed between any 4 independent individuals, independent of their sustained MRD- status.

We agree the permutation test does not show that sharing of TCR β sequences is greater in individuals with sustained MRD-. However, Fig. 5C does not attempt to make this claim. Instead, it shows that we have identified 74 sequences exclusive to individuals with sustained MRD- and these 74 sequences are unlikely to have been shared by random chance. Our evidence for that is 1) the Fisher Exact test comparing the number of patients with and without sustained MRD- that share TCR β sequences is < 0.05 , 2) they are relatively rare sequences in a healthy population, and 3) they have low CDR3B generation probability. In contrast, the shared sequences among patients without sustained MRD- are common in a healthy population and have a high probability of CDR3 β generation. This suggests the sharing of sequences among patients with sustained MRD- may be because of some reason other than they are highly prevalent. In the discussion, we acknowledged that "further investigation would be necessary to determine the antigen specificity of these TCR β sequences."

Patient HLA background

The authors show that sustained MRD- patients do not overlap in their HLA genotype. This does support the hypothesis that HLA background alone is not contributing their TCR overlap – which was my original comment. However, if the sustained MRD- group do not share HLA genotypes, then any shared TCRs would be unlikely to have the same antigen specificity given they would be seeing antigen presented on (structurally distinct) HLA molecules. This TCRB overlap analysis therefore is simply reporting a level of overlap to be expected in a random group of patients of mixed HLA background.

Sufficient material was available for HLA testing on only a subset of patients who also underwent TCR β sequencing. These included one patient with sustained MRD- and eight patients without sustained MRD-. Unfortunately, there is no available material to perform HLA testing on the remaining patients. For this reason, it appeared from supplemental figure 18 that there was no sharing of HLA type among patients with sustained MRD- since only one patient was tested. Since this is misleading, we have removed the previously requested HLA analysis from the manuscript since no meaningful conclusions may be drawn from it.